# GOOD: A Graph Out-of-Distribution Benchmark

**Shurui Gui**,[*]  **Xiner Li**[*],  **Limei Wang**,  **Shuiwang Ji**
Texas A&M University
College Station, TX 77843
{shurui.gui,lxe,limei,sji}@tamu.edu

## Abstract

Out-of-distribution (OOD) learning deals with scenarios in which training and test data follow different distributions. Although general OOD problems have been intensively studied in machine learning, graph OOD is only an emerging area of research. Currently, there lacks a systematic benchmark tailored to graph OOD method evaluation. In this work, we aim at developing an OOD benchmark, known as GOOD, for graphs specifically. We explicitly make distinctions between covariate and concept shifts and design data splits that accurately reflect different shifts. We consider both graph and node prediction tasks as there are key differences in designing shifts. Overall, GOOD contains 11 datasets with 17 domain selections. When combined with covariate, concept, and no shifts, we obtain 51 different splits. We provide performance results on 10 commonly used baseline methods with 10 random runs. This results in 510 dataset-model combinations in total. Our results show significant performance gaps between in-distribution and OOD settings. Our results also shed light on different performance trends between covariate and concept shifts by different methods. Our GOOD benchmark is a growing project and expects to expand in both quantity and variety of resources as the area develops. The GOOD benchmark can be accessed via https://github.com/divelab/GOOD/.

## 1 Introduction

In machine learning, training and test data are commonly assumed to be i.i.d.. Models designed under this assumption may not perform well when the i.i.d. assumption does not hold. The area of out-of-distribution (OOD) learning deals with scenarios in which training and test data follow different distributions. Two commonly studied OOD settings are covariate shift and concept shift. Over the years, multiple OOD methods have been proposed [11, 43, 3, 40, 24]. To facilitate evaluations, several benchmarks have been curated, including DomainBed [16], OoD-Bench [52], and WILDS [23]. Although both general OOD problems and graph analysis [22, 12, 45, 51, 29] have been intensively studied, graph OOD is only an emerging area of research [49, 48, 59, 6]. Some initial attempts have been made to curate graph OOD benchmarks [20, 9]. However, existing benchmarks lack in several aspects, as detailed in Section 2.

**Covariate shift and concept shift.** Distribution shifts can generally be defined as two types; *i.e.*, covariate shift and concept shift (drift) [38, 35, 47]. Formally, in supervised learning, a model is trained to predict an output $Y \in \mathcal{Y}$ given an input $X \in \mathcal{X}$, also known as a covariate variable. The output $Y$ is categorical in classification and continuous in regression problems. In multi-task learning, the output $Y$ becomes a vector, and we consider each task separately. Since the joint distribution $P(Y, X)$ can be written as $P(Y|X)P(X)$, two types of OOD problems are commonly considered, namely covariate and concept shifts. In covariate shift, the input distributions have been shifted between training and test data. Formally, $P^{\text{train}}(X) \neq P^{\text{test}}(X)$ and $P^{\text{train}}(Y|X) = P^{\text{test}}(Y|X)$,

---

[*]Equal contributions

36th Conference on Neural Information Processing Systems (NeurIPS 2022) Track on Datasets and Benchmarks.

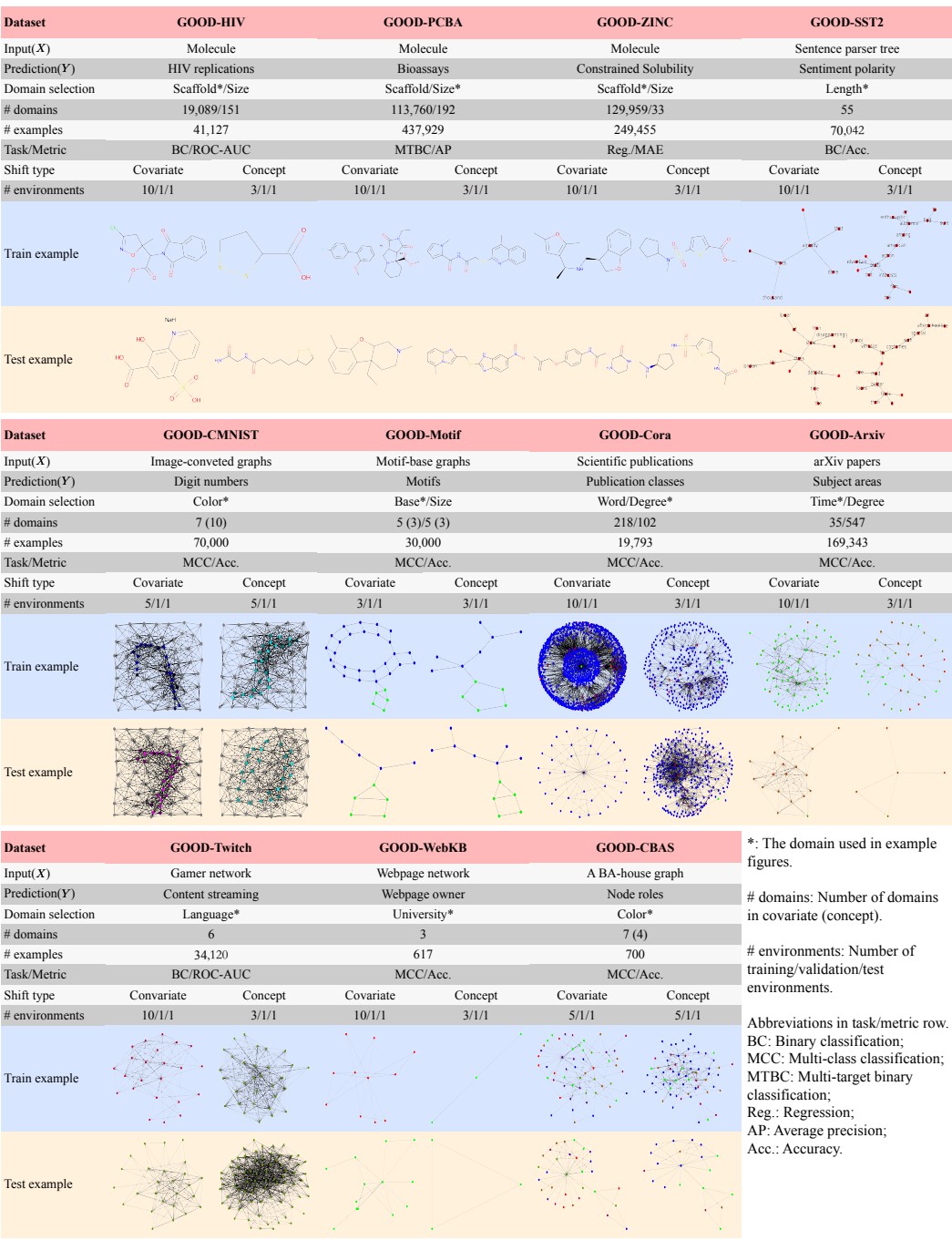

Figure 1: A summary of datasets included in the proposed benchmark. For a covariate shift, the training and test examples are from different domains. For a concept shift, examples are chosen from the same domain with different labels to show the different domain-output correlations.

where $P^{\text{train}}(\cdot)$ and $P^{\text{test}}(\cdot)$ denote training and test distributions, respectively. In contrast, in concept shift, the conditional distribution $P(Y|X)$ has been shifted as $P^{\text{train}}(Y|X) \neq P^{\text{test}}(Y|X)$ and $P^{\text{train}}(X) = P^{\text{test}}(X)$. In this work, we explicitly make distinctions and consider both shifts.

**Differences between graph OOD and general OOD.** Traditional OOD methods commonly focus on simple structure equation models [3, 1, 39, 32] or computer vision tasks [32, 44, 11, 43]. In these tasks, the inputs are variables or image features, denoted as $F$. However, graph data possess the complex

nature of irregularity and connectivity in topology. A graph is commonly represented as an input pair $(F, A)$, where $F$ are node/edge features, and $A$ is the adjacency matrix. Consequently, graph OOD problems focus not only on general feature distribution shifts but also on structure distribution shifts. Graph neural networks are designed based on $F$ and $A$ to pass messages, demonstrating that structures and features carry different perspectives of input information of a graph. The uniqueness of graph data prompts the development of graph-specific OOD methods [49, 48, 6, 59, 10, 26] and calls for graph OOD benchmarks. [2]

**Contributions and novelty.** In this work, we develop a systematic graph OOD benchmark, known as GOOD. As design principles, we strive to (1) create non-trivial performance gaps between training and test data; and (2) provide carefully designed data environments to ensure that the induced distribution shifts are potentially solvable for models. Specifically, GOOD contains 6 graph-level datasets and 5 node-level datasets as shown in Fig. 1. For each dataset, we select one or two types of domains. Given a domain, we generate no-shift, covariate shift, and concept shift splits for ease of comparison among 10 baselines. We summarize our novel contributions as follows. (1) To the best of our knowledge, no existing OOD benchmark provides both covariate and concept shifts comparison for the same domain selection. This is important as it sheds light on the differences between various shifts with proper variable control. For example, from our experiments, DIR [49] performs favorably against other methods in concept shift of GOOD-HIV size split while failing in the corresponding covariate shift. (2) In terms of graph OOD, we are the first benchmark to include not only graph classification, but also graph regression and node-level datasets, which improves the diversity of graph OOD tasks. (3) GOOD provides numerous comparisons for 51 different dataset splits and 10 OOD methods, providing solid baselines for future method developments.

## 2    Related Work

OOD or distribution shift is a longstanding problem in machine learning and artificial intelligence [17, 38, 35, 41]. To address OOD problem substantially, several benchmarks have been curated [16, 52, 18, 23, 57] to evaluate different algorithms [11, 43, 37, 3, 24, 40, 56, 1]. DomainBed [16] is an early OOD benchmark in computer vision. Following DomainBed, OoD-Bench [52] collects datasets and categorizes them into diversity and correlation shifts. WILDS [23] collects real-world data from wild and studies domain generalization and subpopulation shift. Specifically, domain generalization focuses on disjoint training and test domains, while subpopulation shift considers shifts between majority and minority groups, which leads to insufficient training for minority data. With the success of graph neural networks [22, 51, 45, 12, 28, 13, 30, 31], graph OOD problems are gaining growing attention [49, 48, 6, 59, 10, 26, 27, 8]. GDS [9] collects several datasets to compare the performance of well-known baselines and data augmentation methods. DrugOOD [20] is a recent benchmark specifically designed for molecular graph OOD problems. It is curated based on a large-scale bioassay dataset ChEMBL [33] and includes an automated pipeline for obtaining more datasets.

**Differences with existing (graph) OOD benchmarks.** Generalization abilities of OOD algorithms for covariate shift [42] and concept shift [47, 17, 2, 24] differ fundamentally. However, existing benchmarks, including but not limited to graph OOD benchmarks, either ignore one of the shifts or fail to compare the two shifts of the same feature on the same dataset. Firstly, most existing OOD benchmarks include only one type of shift. For example, DrugOOD [20] focuses on domain generalization for molecules, exclusively considering covariate shift. WILDS [23] includes domain generalization and subpopulation shift, which are two cases of covariate shift, while still ignoring concept shift. Secondly, a few benchmarks [52] involve both shifts but simply categorize each dataset as one of these two shifts. GDS [9] collects eight datasets but makes no distinctions between the two shifts; among their datasets, we can categorize ColoredMNIST as concept shift and others as covariate shift. In contrast, our benchmark proposes novel dataset splitting methods to generate both shifts for the same domain selection on the same dataset to enable comparison between shifts. This variable-controlled comparison enables a more thorough analysis of shifts given any specific domain, leading to a more comprehensive OOD benchmark. Furthermore, we curate more diverse graph datasets with diverse tasks, including single/multi-task graph classification, graph regression, and node classification, while neither GDS nor DrugOOD includes graph regression or node-level

---

[2]For simplicity, we use $X$ instead of $(F, A)$ to denote inputs of graphs in our benchmark design. Therefore, a feature in $X$ may refer to not only a node feature but also a specific graph structure.

tasks. In addition, while GDS and DrugOOD do not benchmark any graph-specific OOD methods, we evaluate 4 graph-specific OOD methods, shedding light on further graph OOD research.

## 3   The GOOD Benchmark Design

When training and test samples are assumed to be i.i.d., random split is commonly used to split datasets into training and test sets. In contrast, splits in OOD problems should be carefully designed in order to accurately assess the generalization ability of algorithms. In GOOD, we consider both covariate and concept shifts and meticulously design data splits to ensure these shifts are reflected. Formally, following prior invariant learning work [3, 8, 1, 39, 32], as shown in Fig. 2b, $C, S_1, S_2 \in \mathcal{Z}$ are the latent variables that causes target $Y \in \mathcal{Y}$, is non-causally associated with $Y$, and is independent to $Y$, respectively. $\rightarrow$ denotes the causal mapping. $S_2$ is commonly caused by target-irrelevant environments.[3] In the input feature space, given input features $X \in \mathcal{X}$, we assume the invariant features $X_{\text{inv}} \in X$ are projected by an injective function from $C$.[4] Therefore, $X_{\text{inv}}$ can fully determine $Y$. $X_{\text{ass}}$ denotes input features associated with $Y$ by confounding and anti-causal associations through $C$ and $S_1$. $X_{\text{ind}}$ are input features independent to $Y$ and are only caused by $S_2$. In practice, it might be hard to strictly separate $X_{\text{inv}}$, $X_{\text{ass}}$, and $X_{\text{ind}}$. We try to only select and shift part of $X_{\text{ind}}$, but since our shift splits contain significant dataset shifts, the selection of parts of $X_{\text{ass}}$ won't affect the benchmarking performance. Though, we use $X_{\text{ind}}$ throughout this paper for simplicity.

### 3.1   Covariate shifts

Domain generalization methods follow the covariate shift assumption [5] and assume that the covariate distribution $P(X)$ shifts across splits, while the concept distribution $P(Y|X)$ remains the same. This implies that a shift of $P(X)$ should not cause corresponding shift in $P(Y|X)$. That is, covariate shift can only happen on input features that are not associated with $Y$. Therefore, with prior knowledge, we can manually select and shift one or several of these irrelevant features, $X_{\text{ind}}$, to build covariate splits. Different $X_{\text{ind}}$ feature values indicate different domains, and each domain can be viewed as a split. For instance, in the graph ColoredMNIST dataset in which we distinguish hand-written digits with colors, the color is irrelevant with labels. Thus, in our covariate splits, digits with different colors belong to corresponding color domains, and each domain becomes a split.

Formally, possible values of $X_{\text{ind}}$ are discrete and finite. Therefore, we define each domain by its unique $X_{\text{ind}}$ value, forming its unique input distribution $P(X)$. Then, a dataset can be viewed as a mixture of $|\mathcal{D}|$ domains as $\mathcal{D} = \{d_1, \ldots, d_{|\mathcal{D}|}\}$. For a domain $d_i$, we represent its input distribution as $P_{d_i}(X)$. Specifically, $P_{d_i}(X_{\text{inv}}, X_{\text{ass}})$ is fixed while $P_{d_i}(X_{\text{ind}} = \mathbf{x}_i) = 1$, where $\mathbf{x}_i$ is the value of $X_{\text{ind}}$ in domain $d_i$. Since $X_{\text{ind}}$ is independent to $Y$, $P_{d_i}(Y|X) = P(Y|X)$. The data distribution is

$$P(Y, X) = \sum_{i=1}^{|\mathcal{D}|} w_i P_{d_i}(Y, X) = \sum_{i=1}^{|\mathcal{D}|} w_i P_{d_i}(X) P(Y|X), \tag{1}$$

where $w_i$ is the mixture weight for domain $d_i$.

**Comparison of covariate split design on 3 types of datasets.** We perform covariate shift splits on synthetic, semi-artificial, and real-world datasets. For a synthetic dataset, given the $X_{\text{ind}}$ feature values of each domain, we generate graphs according to its domain distributions, respectively. For a semi-artificial dataset, given a graph, we generate extra variant features to produce modified graphs that follow the domain distribution. Since we cannot create or modify graphs for real-world datasets, we use carefully designed data splits. As shown in Fig. 2a, we sort the graphs by their domain $d_i \in \mathcal{D}$ and then divide the dataset into five domain splits with a specific split ratio, *e.g.*, $20\%$, for each domain. Finally, the training, validation, and test sets are obtained based on domains without intersections. The difference between synthetic/semi-artificial datasets and real-world datasets is that the domain of graphs in artificial datasets can be defined arbitrarily before feature modifications, but the domain of graphs in real-world datasets should be defined strictly according to its features.

---

[3]We do not explicitly consider unobserved confounders in this paper.

[4]This assumption is for clearer input distinction, and will not introduce any side effect to further analysis, since the shift split design focuses on selecting $X_{\text{ind}}$ instead of distinguishing $X_{\text{inv}}$ and $X_{\text{ass}}$.

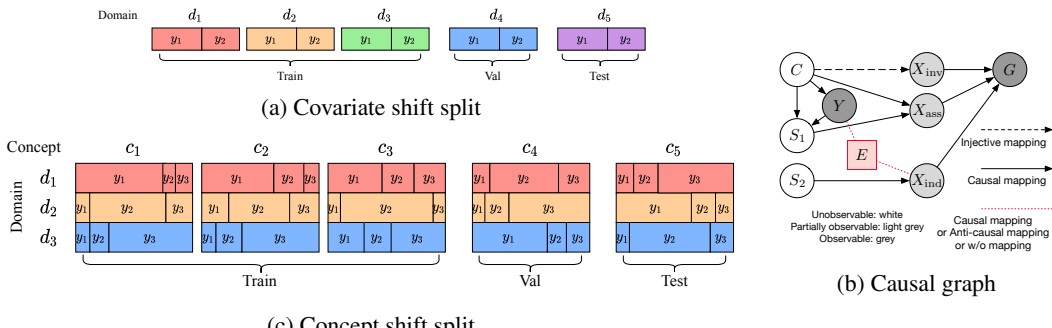

(a) Covariate shift split

(c) Concept shift split

(b) Causal graph

Figure 2: (a) Illustration of covariate shift split. Five domains are denoted as different colors, where each domain includes outputs of the same distribution. We sort the dataset according to the domain $d_i$, then group them into train/validation/test sets. (c) Illustration of concept shift split. Each concept includes all three domains, and each domain has spurious correlations with a specific output in a concept. For example, in concept $c_1$, the domain colored in red is highly associated with $y_1$, but this domain corresponds to $y_2$ in concept $c_4$. Note that the distributions of concepts in training are similar. (b) Illustration of causal graph [36, 37] for dataset generation and observation. (Left) $C$, $S_1$, $S_2$ locate in the latent space and are not observable. (Middle) $X_{\text{inv}}$, $X_{\text{ass}}$, and $X_{\text{ind}}$ are input features that can be partially observed and selected manually, such as motif shapes or molecule scaffolds. (Right) $G$ is the graph data input including node features and adjacency matrices. $E \in \mathcal{E}$ is the environment variable that can determine or be determined by $X_{\text{inv}}$ and $Y$ according to different types of datasets and shifts. Detailed discussions can be found in Appendix A.

## 3.2 Concept shifts

In contrast to covariate shift, concept shift considers the scenario in which the concept distribution $P(Y|X)$ is shifted across splits. Since $X_{\text{inv}}$ can fully determine $Y$, $P(Y|X_{\text{inv}})$ is invariant. Thus, shifts of $P(Y|X)$ can only happen with shifts of $P(Y|X_{\text{ass}})$ and $P(Y|X_{\text{ind}})$. Since $X_{\text{ind}}$ is irrelevant with $Y$, the correlation $P(Y|X_{\text{ind}})$ between $Y$ and $X_{\text{ind}}$ in each domain is spurious correlation. However, the association built between $Y$ and $X_{\text{ind}}$ will also connect $X_{\text{inv}}$ to $X_{\text{ind}}$ through $Y$ according to Fig. 2b, leading to the change of $P(X_{\text{inv}}|X_{\text{ind}})$, so that $P(X)$ will be changed inevitably. Therefore, we can only build major concept shifts with necessary covariate shifts. We will still call it concept shift for simplicity and distinction. Therefore, given the selected domain features $X_{\text{ind}}$, we can build concept shift splits by manually creating such spurious correlations of certain rates. For example, the spurious correlation rate between the domain $d_i$ and the output value $y_j$ can be set as $P(Y = y_j|X_{\text{ind}} = \mathbf{x}_i) = 90\%$. Specifically, different spurious correlation rates define different concepts, and each concept can be viewed as a split. We use the graph ColoredMNIST dataset as an example, as shown in Fig. 2c. In every concept, each color domain is highly correlated with a label. Therefore, in our concept splits, different spurious color-label correlation rates determine different concepts, and each concept becomes a split.

Formally, a dataset can be viewed as a mixture of $|\mathcal{C}|$ concepts $\mathcal{C} = \{c_1, \ldots, c_{|\mathcal{C}|}\}$ in our concept shift split. We use $P_{y_j,d_i}(Y)$ to represent a certain output distribution on value $y_j$ given domain $d_i$, defined as $P(Y = y_j|X_{\text{ind}} = \mathbf{x}_i) = 1$. We first consider the classification case in which $Y$ is categorical. Given a concept $c_k$, we formulate its 2-D conditional distribution $P_{c_k}(Y|X)$ by describing multiple 1-D distributions; that is, for each domain $d_i$,

$$P_{c_k}(Y|X_{\text{ind}} = \mathbf{x_i}) = \sum_{j=1}^{|\mathcal{Y}|} q_{i,j}^k P_{y_j,d_i}(Y), \qquad (2)$$

where $q_{i,j}^k$ is the rate of the spurious correlation in concept $c_k$ between domain $d_i$ and output $y_j$. In the regression case where $Y$ is continuous, the sum becomes integral. In the multi-task case, $Y$ and $y_j$ become vectors. The overall dataset distribution can be written as

$$P(Y, X) = \sum_{k=1}^{|\mathcal{C}|} w_k P_{c_k}(Y, X) = \sum_{k=1}^{|\mathcal{C}|} w_k P(X)_{c_k} P_{c_k}(Y|X), \qquad (3)$$

where $w_k$ is the mixture weight for concept $c_k$.

**Comparisons of concept split design on 3 types of datasets.** In practice, we create significant concept shifts between training, validation, and test sets, in which the domain-output correlations are completely different. Note that mixing different domain-output correlations can weaken the spuriousness of these correlations. Thus, concepts within the training set are designed to have similar domain-output correlations to guarantee the concept shift between training and test. Concretely, we perform concept shift splits on synthetic, semi-artificial, and real-world datasets. For synthetic datasets, we generate graphs where the domain feature is highly correlated with a specific output according to the preset correlation in the concept. For semi-artificial datasets, given a graph and a concept, we generate extra features as domains to build spurious domain-output correlations. For real-world datasets, we cannot create or modify data. Thus, we propose a screening approach to scan and select graphs in the dataset. Each graph has a probability to be included in a concept $c_k$ according to the value of $q_{i,j}^k$. To conclude, the difference in concept splits between artificial datasets and real-world datasets is located in whether we can *arbitrarily* determine the concept of a graph. We can define the concept of a graph in artificial datasets, but the concept of a graph in real-world datasets is defined with a probability according to $X_{\text{ind}}$ and target $Y$.

### 3.3 Environments

Many current OOD learning algorithms [40, 24] follow the framework of invariant causal predictor (ICP) [37] and invariant risk minimization (IRM) [3], assuming that the training data form groups by distributions, known as environments. This framework assumes the data are similar within an environment and dissimilar across different environments. Since OOD problems are complicated and multi-perspective, works under this framework explicitly or implicitly injects environment information to models to figure out the specific generalization direction for better OOD performances. Specifically, the shift between training and test data, though more significant, should be similarly reflected among different training environments, so that OOD models can potentially grasp the shift between training and test data by learning the shifts among different training environments. Following this strategy, to enhance the OOD generalization ability of models, we use the distribution shift information provided by the difference of training environments to convey the types of shifts expected between training and test data. In covariate shift, environments take the form of domains. During training, models can learn from $P^{\text{train}}(X_{\text{ind}})$, which varies across domains, that $X_{\text{ind}}$ is not causally related to labels, thereby preventing the unknown $P^{\text{test}}(X_{\text{ind}})$ from misleading predictions during test. In concept shifts, environments take the form of concepts. By learning from different spurious correlations across training concepts, models can learn that the domain-output correlations $P^{\text{train}}(Y|X_{\text{ind}})$ are spurious, thereby avoiding being misled by the new spurious correlation $P^{\text{test}}(Y|X_{\text{ind}})$ during test.

Formally, we consider a dataset with a set of $|\mathcal{E}|$ environments $\mathcal{E} = \{e_1, \ldots, e_{|\mathcal{E}|}\}$, each with distribution $P_e(Y, X)$ for $e \in \mathcal{E}$ (Fig 2b). In this case, the dataset distribution $P(Y, X) = \sum_e P_e(Y, X)$. Specifically, for both covariate and concept shifts, the distributions $P^{\text{train}}$ and $P^{\text{test}}$ are weighted combinations of environment distributions $P_e(Y, X)$. With the training and test environments $\mathcal{E}^{\text{train}}, \mathcal{E}^{\text{test}} \subset \mathcal{E}$, we express $P^{\text{train}} = \sum_{e \in \mathcal{E}^{\text{train}}} w_e^{\text{train}} P_e(Y, X)$ and $P^{\text{test}} = \sum_{e \in \mathcal{E}^{\text{test}}} w_e^{\text{test}} P_e(Y, X)$, where $w_e^{\text{train}}$ and $w_e^{\text{test}}$ are the weights for each training and test environment, respectively.

## 4 The GOOD Datasets

In this section, we introduce the datasets in GOOD. The benchmark contains 11 datasets, covering multiple tasks and data sources. For each dataset, we select one or two domain features. Then we apply covariate and concept shift splits per domain to create diverse distribution shifts between training, OOD validation, and OOD test sets. Finally, we shuffle the training set and divide it into final training set, in-domain (ID) validation set, and in-domain (ID) test set. Summary statistics of datasets are given in Fig. 1. Other details and data processing details are included in Appendix A.

### 4.1 Graph prediction tasks

**GOOD-HIV** is a small-scale real-world molecular dataset adapted from MoleculeNet [50]. The inputs are molecular graphs in which nodes are atoms, and edges are chemical bonds. The task is to predict whether the molecule can inhibit HIV replication. We design splits based on two domain selections, namely, scaffold and size. The first one is Bemis-Murcko scaffold [4] which

is the two-dimensional structural base of a molecule. The second one is the number of nodes in a molecular graph, an inevitable structural feature of a graph. Both features should not determine the label, therefore, both can become major sources of distribution shifts. For each domain selection, the value space for the feature is very large, therefore we cluster graphs with similar domain values into one environment, improving the OOD learning procedure and reducing the training time complexity.

**GOOD-PCBA** is a real-world molecular dataset from Wu et al. [50]. It includes 128 bioassays, forming 128 binary classification tasks. Due to the extremely unbalanced classes (only 1.4% positive labels), we use the Average Precision (AP) averaged over the tasks as the evaluation metric. GOOD-PCBA uses the same domain selections as GOOD-HIV.

**GOOD-ZINC** is a real-world molecular property regression dataset from ZINC database [15]. The inputs are molecular graphs with up to 38 heavy atoms, and the task is to predict the constrained solubility [21, 25] of molecules. GOOD-ZINC uses the same domain selections as GOOD-HIV.

**GOOD-SST2** is a real-world natural language sentimental analysis dataset adapted from Yuan et al. [54]. Each sentence is transformed into a grammar tree graph, where each node represents a word with corresponding word embeddings as node features. The dataset forms a binary classification task to predict the sentiment polarity of a sentence. We select sentence lengths as domains since the length of a sentence should not affect the sentimental polarity.

**GOOD-CMNIST** is a semi-artificial dataset designed for node feature shifts. It contains graphs of hand-written digits transformed from MNIST database using superpixel techniques [34]. Following Arjovsky et al. [3], we color digits according to their domains and concepts. Specifically, in covariate shift split, we color digits with 7 different colors, and digits with the first 5 colors, the 6th color, and the 7th color are categorized into training, validation, and test sets. In concept shift split, we color digits with 10 colors. Each color is highly correlated with one digit label in the training set, while colors have weak correlations and no correlation with labels in validation and test sets, respectively.

**GOOD-Motif** is a synthetic dataset motivated by Spurious-Motif [49] and is designed for structure shifts. Particularly, GOOD-CMNIST and GOOD-Motif compose an OOD algorithm check for both feature/structure shifts. Each graph in the dataset is generated by connecting a base graph and a motif, and the label is determined by the motif solely. Instead of combining the base-label spurious correlations and size covariate shift together as in Wu et al. [49], we study covariate and concept shifts separately. Specifically, we generate graphs using five label irrelevant base graphs (wheel, tree, ladder, star, and path) and three label determining motifs (house, cycle, and crane). To create covariate and concept splits, we select the base graph type and the size as domain features.

## 4.2 Node prediction tasks

**GOOD-Cora** is a citation network adapted from the full Cora dataset [7]. The input is a small-scale citation network graph, in which nodes represent scientific publications and edges are citation links. The task is a 70-class classification of publication types. We generate splits based on two domain selections, namely, word and degree. The first one is the word diversity defined by the selected-word-count of a publication, purely irrelevant with the label. The second one is node degree in the graph, implying that the popularity of a paper should not determine the class of a paper.

**GOOD-Arxiv** is a citation dataset adapted from OGB [19]. The input is a directed graph representing the citation network among the computer science (CS) arXiv papers. Nodes in the graph represent arXiv papers, and directed edges represent citations. The task is predicting the subject area of arXiv CS papers, forming a 40-class classification problem. We generate splits based on two domain selections; *i.e.*, time (publication year) and node degree.

**GOOD-Twitch** is a gamer network dataset. The nodes represent gamers with games as node features, and the edge represents the friendship connection of gamers. The binary classification task is to predict whether a user streams mature content. The domain of GOOD-Twitch splits includes user language, implying that the prediction target should not be biased by the language a user uses.

**GOOD-WebKB** is a university webpage network dataset. A node in the network represents a webpage, with words appearing in the webpage as node features, and edges are hyperlinks between webpages. Its 5-class prediction task is to predict the classes of webpages. We split GOOD-WebKB according to the domain university, suggesting that classified webpages are based on word contents and link connections instead of university features.

Table 1: ID and OOD performance gaps learned with ERM across 51 splits. The metric and domain selections for each dataset are in Fig. 1. ↑ indicates higher values correspond to better performance while ↓ indicates lower for better. ID test results with ID validations are denoted as $\text{ID}_{\text{ID}}$, while OOD test results with ID/OOD validations are written as $\text{OOD}_{\text{ID}}$ and $\text{OOD}_{\text{OOD}}$, respectively. Note that the no-shift random split only has the ID setting. We report the average values over 10 runs. The standard deviations are listed in Appendix D.

| | domain selection 1 | | | | | | | domain selection 2 | | | | | | |
| | covariate | | | concept | | | no-shift | covariate | | | concept | | | no-shift |
| | $\text{ID}_{\text{ID}}$ | $\text{OOD}_{\text{ID}}$ | $\text{OOD}_{\text{OOD}}$ | $\text{ID}_{\text{ID}}$ | $\text{OOD}_{\text{ID}}$ | $\text{OOD}_{\text{OOD}}$ | $\text{ID}_{\text{ID}}$ | $\text{ID}_{\text{ID}}$ | $\text{OOD}_{\text{ID}}$ | $\text{OOD}_{\text{OOD}}$ | $\text{ID}_{\text{ID}}$ | $\text{OOD}_{\text{ID}}$ | $\text{OOD}_{\text{OOD}}$ | $\text{ID}_{\text{ID}}$ |
|---|---|---|---|---|---|---|---|---|---|---|---|---|---|---|
| GOOD-HIV↑ | 82.79 | 68.86 | 69.58 | 84.22 | 65.31 | 72.33 | 80.86 | 83.72 | 58.41 | 59.94 | 88.05 | 44.75 | 63.26 | 80.86 |
| GOOD-PCBA↑ | 33.45 | 16.87 | 16.89 | 25.95 | 21.34 | 21.63 | 33.77 | 34.31 | 17.81 | 17.86 | 32.54 | 14.83 | 15.36 | 33.77 |
| GOOD-ZINC↓ | 0.1224 | 0.1825 | 0.1995 | 0.1222 | 0.1328 | 0.1306 | 0.1233 | 0.1199 | 0.2569 | 0.2427 | 0.1315 | 0.1418 | 0.1403 | 0.1233 |
| GOOD-SST2↑ | 89.82 | 77.76 | 81.30 | 94.43 | 67.26 | 72.43 | 91.61 | – | – | – | – | – | – | – |
| GOOD-CMNIST↑ | 77.96 | 26.90 | 28.60 | 90.00 | 40.80 | 42.87 | 77.30 | – | – | – | – | – | – | – |
| GOOD-Motif↑ | 92.60 | 69.97 | 68.66 | 92.02 | 80.87 | 81.44 | 92.09 | 92.28 | 51.28 | 51.74 | 91.73 | 69.41 | 70.75 | 92.09 |
| GOOD-Cora↑ | 70.43 | 64.44 | 64.86 | 66.05 | 64.20 | 64.60 | 69.41 | 72.27 | 55.76 | 56.30 | 68.71 | 60.38 | 60.54 | 69.42 |
| GOOD-Arxiv↑ | 72.69 | 70.64 | 71.08 | 74.76 | 65.70 | 67.32 | 73.02 | 77.47 | 58.53 | 58.91 | 75.27 | 61.77 | 62.99 | 72.99 |
| GOOD-Twitch↑ | 70.66 | 47.73 | 48.95 | 80.29 | 48.57 | 57.32 | 68.05 | – | – | – | – | – | – | – |
| GOOD-WebKB↑ | 38.25 | 11.64 | 14.29 | 65.00 | 24.77 | 27.83 | 47.85 | – | – | – | – | – | – | – |
| GOOD-CBAS↑ | 89.29 | 77.57 | 76.00 | 89.79 | 82.22 | 82.36 | 99.43 | – | – | – | – | – | – | – |

**GOOD-CBAS** is a synthetic dataset modified from BA-Shapes [53]. The input is a graph created by attaching 80 house-like motifs to a 300-node Barabási–Albert base graph, and the task is to predict the role of nodes, including the top/middle/bottom node of a house-like motif or the node from the base graph, forming a 4-class classification task. Instead of using constant node features, we generate colored features as in GOOD-CMNIST so that OOD algorithms need to tackle node color differences in covariate splits and color-label correlations in concept splits.

## 5 Experimental Studies

We conduct experiments on 11 datasets with 10 baseline methods. For each dataset, we use the same GNN backbone for all baseline methods for fair comparisons. Specifically, we use GIN-Virtual [51, 14] and GCN [22, 55] as GNN backbones for graph and node prediction tasks, respectively. Note that for GOOD-Motif, we adopt GIN [51] as the GNN backbone since adding virtual nodes does not improve the performance. For all experiments, we select the best checkpoints for ID and OOD tests according to results on ID and OOD validation sets, respectively. Experimental details and hyper-parameter selections are provided in Appendix B. All the datasets, implementation codes, and best checkpoints to reproduce the results in this paper are available at https://github.com/divelab/GOOD/.

### 5.1 In-distribution versus out-of-distribution performance gap

As introduced in Sec. 1, one principle for designing GOOD is to create non-trivial distribution shifts and performance gaps between training and test data. Equivalently, we expect distinct performance gaps between ID and OOD settings. To verify performance gaps, we run experiments using empirical risk minimization (ERM) and summarize the results in Table 1. The differences between $\text{ID}_{\text{ID}}$ and $\text{OOD}_{\text{ID}}$ or $\text{OOD}_{\text{OOD}}$ for each domain selection and distribution shift show the substantial and consistent performance gap between the ID and OOD settings. In addition, for most splits, $\text{OOD}_{\text{OOD}}$ is better than $\text{OOD}_{\text{ID}}$. This implies that OOD validation sets outperform ID validation sets in selecting models with better generalization ability.

### 5.2 Performance of baseline algorithms

In our benchmark, we conduct experiments with 10 baseline methods. Based on the comparison results, we provide an analysis of the learning strategy of OOD methods.

#### 5.2.1 Baseline methods

We consider empirical risk minimization (ERM) and 9 OOD algorithms as baselines, among which 4 are graph-specific methods. Firstly, we choose two domain adaptation algorithms that target minimizing feature discrepancies. DANN [11] adversarially trains the regular classifier and a domain classifier to make features indistinguishable. Deep Coral [43] encourages features in different domains to be similar by minimizing the deviation of covariant matrices from different domains. Furthermore,

Table 2: $ID_{ID}$ and $OOD_{OOD}$ performances of 10 baselines on 11 datasets. All numerical results are averages across 3 to 10 random runs. Numbers in **bold** represent the best results. OOM denotes out of memory. Additional results are in Appendix D. More empirical results and analysis are in Appendix C.

| covariate | GOOD-HIV↑ | | | | GOOD-PCBA↑ | | | | GOOD-ZINC↓ | | | | GOOD-CMNIST↑ | | |
|---|---|---|---|---|---|---|---|---|---|---|---|---|---|---|---|
| | scaffold | | size | | scaffold | | size | | scaffold | | size | | color | | |
| | $ID_{ID}$ | $OOD_{OOD}$ | $ID_{ID}$ | $OOD_{OOD}$ | $ID_{ID}$ | $OOD_{OOD}$ | $ID_{ID}$ | $OOD_{OOD}$ | $ID_{ID}$ | $OOD_{OOD}$ | $ID_{ID}$ | $OOD_{OOD}$ | $ID_{ID}$ | $OOD_{OOD}$ | |
| ERM | **82.79** | 69.58 | 83.72 | 59.94 | 33.45 | 16.89 | 34.31 | 17.86 | 0.1224 | 0.1995 | 0.1199 | 0.2427 | 77.96 | 28.60 | graph |
| IRM | 81.35 | 67.97 | 81.33 | 59.00 | 33.56 | 16.90 | 34.28 | **18.05** | 0.1213 | 0.2025 | 0.1222 | 0.2403 | 77.92 | 27.83 | |
| VREx | 82.11 | **70.77** | 83.47 | 58.53 | **33.88** | **16.98** | 34.09 | 17.79 | 0.1211 | 0.2094 | 0.1234 | **0.2384** | 77.98 | 28.48 | |
| GroupDRO | 82.60 | 70.64 | 83.79 | 58.98 | 33.81 | 16.98 | 33.95 | 17.59 | **0.1168** | **0.1934** | 0.1180 | 0.2423 | 77.98 | 29.07 | |
| DANN | 81.18 | 70.63 | 83.90 | 58.68 | 33.63 | 16.90 | 34.17 | 17.86 | 0.1186 | 0.2004 | 0.1188 | 0.2439 | 78.00 | **29.14** | |
| Deep Coral | 82.53 | 68.61 | **84.70** | **60.11** | 33.47 | 16.93 | **34.49** | 17.94 | 0.1185 | 0.2036 | **0.1134** | 0.2505 | **78.64** | 29.05 | |
| Mixup | 82.29 | 68.88 | 83.16 | 59.03 | 30.22 | 16.59 | 30.63 | 17.06 | 0.1279 | 0.2240 | 0.1255 | 0.2748 | 77.40 | 26.47 | |
| DIR | 82.54 | 67.47 | 80.46 | 57.11 | 32.55 | 14.98 | 32.89 | 16.61 | 0.3799 | 0.6493 | 0.1541 | 0.5482 | 31.09 | 20.60 | |

| covariate | GOOD-Motif↑ | | | | GOOD-SST2↑ | | concept | GOOD-Motif↑ | | | | GOOD-SST2↑ | | |
|---|---|---|---|---|---|---|---|---|---|---|---|---|---|---|
| | base | | size | | length | | | base | | size | | length | | |
| | $ID_{ID}$ | $OOD_{OOD}$ | $ID_{ID}$ | $OOD_{OOD}$ | $ID_{ID}$ | $OOD_{OOD}$ | | $ID_{ID}$ | $OOD_{OOD}$ | $ID_{ID}$ | $OOD_{OOD}$ | $ID_{ID}$ | $OOD_{OOD}$ | |
| ERM | 92.60 | 68.66 | 92.28 | 51.74 | **89.82** | 81.30 | ERM | 92.02 | 81.44 | 91.73 | **70.75** | **94.43** | 72.43 | graph |
| IRM | 92.60 | 70.65 | 92.18 | 51.41 | 89.41 | 79.91 | IRM | 92.00 | 80.71 | 91.68 | 69.77 | 94.10 | **77.47** | |
| VREx | 92.60 | **71.47** | 92.25 | **52.67** | 89.51 | 80.64 | VREx | **92.05** | **81.56** | 91.67 | 70.24 | 94.26 | 73.16 | |
| GroupDRO | 92.61 | 68.24 | **92.29** | 51.95 | 89.59 | **81.35** | GroupDRO | 92.01 | 81.43 | 91.67 | 69.98 | 94.41 | 71.86 | |
| DANN | 92.60 | 65.47 | 92.23 | 51.46 | 89.60 | 79.71 | DANN | 92.02 | 81.33 | **91.81** | 70.72 | 94.02 | 76.03 | |
| Deep Coral | 92.61 | 68.88 | 92.22 | 50.97 | 89.68 | 79.81 | Deep Coral | 92.01 | 81.37 | 91.68 | 70.49 | 94.25 | 72.34 | |
| Mixup | **92.68** | 70.08 | 92.02 | 51.48 | 89.78 | 80.88 | Mixup | 91.89 | 77.63 | 91.45 | 67.81 | 94.12 | 73.34 | |
| DIR | 87.73 | 61.50 | 84.53 | 50.41 | 84.30 | 77.65 | DIR | 91.60 | 72.14 | 73.10 | 56.28 | 93.71 | 68.76 | |

| concept | GOOD-HIV↑ | | | | GOOD-PCBA↑ | | | | GOOD-ZINC↓ | | | | GOOD-CMNIST↑ | | |
|---|---|---|---|---|---|---|---|---|---|---|---|---|---|---|---|
| | scaffold | | size | | scaffold | | size | | scaffold | | size | | color | | |
| | $ID_{ID}$ | $OOD_{OOD}$ | $ID_{ID}$ | $OOD_{OOD}$ | $ID_{ID}$ | $OOD_{OOD}$ | $ID_{ID}$ | $OOD_{OOD}$ | $ID_{ID}$ | $OOD_{OOD}$ | $ID_{ID}$ | $OOD_{OOD}$ | $ID_{ID}$ | $OOD_{OOD}$ | |
| ERM | 84.22 | 72.33 | 88.05 | 63.26 | 25.95 | 21.63 | 32.54 | 15.36 | 0.1222 | 0.1306 | 0.1315 | 0.1403 | 90.00 | 42.87 | graph |
| IRM | 82.89 | 72.59 | **88.62** | 59.90 | 25.89 | 21.22 | 32.99 | 16.07 | 0.1225 | 0.1314 | 0.1278 | 0.1368 | 90.02 | 42.80 | |
| VREx | 83.84 | 72.60 | 88.28 | 60.23 | **26.62** | 22.02 | 32.49 | 15.59 | 0.1186 | 0.1270 | 0.1309 | 0.1419 | 89.99 | 43.31 | |
| GroupDRO | 83.40 | **73.64** | 88.28 | 61.37 | 26.32 | 21.83 | **33.03** | 15.99 | 0.1207 | 0.1281 | **0.1251** | 0.1369 | **90.02** | **43.32** | |
| DANN | 83.87 | 71.92 | 87.88 | 65.27 | 26.07 | 21.64 | 32.74 | 15.78 | **0.1172** | **0.1256** | 0.1253 | **0.1339** | 89.94 | 43.11 | |
| Deep Coral | **84.65** | 72.97 | 87.88 | 62.28 | 26.38 | 21.95 | 32.67 | 16.20 | 0.1187 | 0.1279 | 0.1287 | 0.1370 | 89.94 | 43.16 | |
| Mixup | 82.36 | 72.03 | 87.64 | 64.87 | 23.73 | 19.78 | 30.23 | 13.36 | 0.1353 | 0.1475 | 0.1423 | 0.1522 | 89.95 | 40.96 | |
| DIR | 83.28 | 69.05 | 79.19 | **72.61** | 25.85 | **22.20** | 30.53 | **16.86** | 0.3501 | 0.3865 | 0.2348 | 0.2871 | 86.76 | 22.69 | |

| covariate | GOOD-Cora↑ | | | | GOOD-Arxiv↑ | | | | GOOD-CBAS↑ | | GOOD-Twitch↑ | | GOOD-WebKB↑ | | |
|---|---|---|---|---|---|---|---|---|---|---|---|---|---|---|---|
| | word | | degree | | time | | degree | | color | | language | | university | | |
| | $ID_{ID}$ | $OOD_{OOD}$ | $ID_{ID}$ | $OOD_{OOD}$ | $ID_{ID}$ | $OOD_{OOD}$ | $ID_{ID}$ | $OOD_{OOD}$ | $ID_{ID}$ | $OOD_{OOD}$ | $ID_{ID}$ | $OOD_{OOD}$ | $ID_{ID}$ | $OOD_{OOD}$ | |
| ERM | 70.43 | 64.86 | 72.27 | 56.30 | 72.69 | 71.08 | 77.47 | 58.91 | 89.29 | 76.00 | 70.66 | 48.95 | 38.25 | 14.29 | node |
| IRM | 70.27 | 64.77 | 72.64 | 56.28 | 72.66 | 71.04 | 77.50 | 58.98 | 91.00 | 76.00 | 69.75 | 47.21 | 39.34 | 13.49 | |
| VREx | 70.47 | 64.80 | 72.25 | 56.30 | 72.66 | 71.12 | 77.49 | 58.99 | **91.14** | 77.14 | 70.66 | 48.99 | 39.34 | 14.29 | |
| GroupDRO | 70.41 | 64.72 | 72.18 | 56.29 | 72.68 | 71.15 | 77.46 | **59.08** | 90.86 | 76.14 | 70.84 | 47.20 | 39.89 | 17.20 | |
| DANN | 70.66 | 64.77 | 72.47 | 56.10 | **72.74** | 71.05 | 77.51 | 59.00 | 90.14 | **77.57** | 70.67 | 48.98 | 39.89 | 15.08 | |
| Deep Coral | 70.47 | 64.72 | 72.16 | 56.35 | 72.66 | 71.07 | 77.48 | 58.97 | 91.14 | 75.86 | 70.67 | 49.64 | 38.25 | 13.76 | |
| Mixup | 71.54 | **65.23** | 74.57 | 58.20 | 72.49 | **71.34** | **77.61** | 57.60 | 73.57 | 70.57 | 71.30 | **52.27** | **54.65** | 17.46 | |
| EERM | 68.79 | 61.98 | 73.32 | 56.88 | OOM | OOM | OOM | OOM | 67.62 | 52.86 | **73.87** | 51.34 | 46.99 | **24.61** | |
| SRGNN | 70.27 | 64.66 | 71.37 | 54.78 | 72.50 | 70.83 | 75.96 | 57.52 | 77.62 | 74.29 | 70.58 | 47.30 | 39.89 | 13.23 | |

| concept | GOOD-Cora↑ | | | | GOOD-Arxiv↑ | | | | GOOD-CBAS↑ | | GOOD-Twitch↑ | | GOOD-WebKB↑ | | |
|---|---|---|---|---|---|---|---|---|---|---|---|---|---|---|---|
| | word | | degree | | time | | degree | | color | | language | | university | | |
| | $ID_{ID}$ | $OOD_{OOD}$ | $ID_{ID}$ | $OOD_{OOD}$ | $ID_{ID}$ | $OOD_{OOD}$ | $ID_{ID}$ | $OOD_{OOD}$ | $ID_{ID}$ | $OOD_{OOD}$ | $ID_{ID}$ | $OOD_{OOD}$ | $ID_{ID}$ | $OOD_{OOD}$ | |
| ERM | 66.05 | 64.60 | 68.71 | 60.54 | 74.76 | 67.32 | **75.27** | 62.99 | 89.79 | 82.36 | 80.29 | 57.32 | 65.00 | 27.83 | node |
| IRM | 66.09 | 64.60 | 68.58 | 61.23 | 74.67 | 67.41 | 75.23 | 62.97 | 90.71 | **83.21** | 77.05 | 59.17 | 65.56 | 27.52 | |
| VREx | 66.00 | 64.57 | 68.45 | 60.58 | 74.80 | 67.37 | 75.19 | **63.00** | 89.50 | 82.86 | 80.29 | 57.37 | 65.00 | 27.83 | |
| GroupDRO | 66.17 | **64.62** | 68.37 | 60.65 | 74.73 | **67.45** | 75.19 | 62.88 | 90.36 | 82.00 | 81.95 | **60.27** | 65.00 | 28.14 | |
| DANN | 66.16 | 64.51 | 68.08 | 60.78 | 74.73 | 67.28 | 75.25 | 62.91 | 89.93 | 82.50 | 80.28 | 57.46 | 65.00 | 26.91 | |
| Deep Coral | 66.13 | 64.58 | 68.38 | 60.58 | 74.77 | 67.42 | 75.16 | 62.85 | 89.86 | 82.64 | 80.14 | 56.97 | 65.00 | 28.75 | |
| Mixup | **69.66** | 64.44 | **70.32** | **63.65** | **74.92** | 64.84 | 72.75 | 61.28 | **93.64** | 64.57 | 78.89 | 55.28 | **67.22** | **31.19** | |
| EERM | 65.75 | 63.09 | 66.50 | 58.38 | OOM | OOM | OOM | OOM | 78.33 | 64.29 | **83.91** | 51.94 | 61.67 | 27.83 | |
| SRGNN | 66.45 | **64.62** | 68.34 | 61.08 | 74.64 | 67.17 | 74.83 | 62.09 | 88.57 | 81.43 | 80.21 | 56.05 | 61.67 | 27.52 | |

we adopt two invariant learning baselines following the invariant prediction assumption [37]. IRM [3] searches for data representations that perform well across all environments by penalizing feature distributions that have different optimal linear classifiers for each environment. VREx [24] targets both covariate robustness and the invariant prediction. It specifically reduces the variance of risks in test environments by minimizing the risk variances of training environments. By applying fair optimization, GroupDRO [40] tackles the problem that the distribution minority lacks sufficient training. This method, known as risk interpolation [24], is achieved by explicitly minimizing the loss in the worst training environment. To evaluate the performance of current OOD methods specifically designed for graphs, we include the following 4 graph OOD methods. We incorporate the data augmentation method Mixup [56] following the implementation of Mixup-For-GraphWang et al. [46] designed for graph data, which improves model generalization abilities. DIR [49] selects a subset of graph representations as causal rationales and conducts interventional data augmentation to create

multiple distributions. EERM [48] tries to generate environments by a REINFORCE algorithm to maximize loss variance between environments, while its main loss minimization requires adversarially minimizing loss variance. SRGNN [59] aims at converting the biased training data to the given unbiased distribution, performed through a central moment discrepancy regularizer and the kernel mean matching technique by solving a quadratic problem.

### 5.2.2 Quantitative comparison and analysis

Table 2 shows the $OOD_{OOD}$ and $ID_{ID}$ results of 10 baselines for all splits. Most OOD algorithms have comparable performances with ERM, while many OOD algorithms outperform ERM with certain patterns. Specifically, we observe that the risk interpolation (GroupDRO) and extrapolation (VREx) perform favorably against other methods on multiple datasets and shift splits. VREx outperforms other methods on 7 out of 34 OOD splits, evidencing its learning invariance and robustness, especially for covariate shifts in graph prediction tasks. GroupDRO outperforms 8 out of 34 OOD splits, showing its advantage in fair optimization. The two feature discrepancy minimization methods, DANN and Deep Coral, do not perform well enough. DANN outperforms on 4 splits, and it is especially suitable for graph concept shift splits. Deep Coral outperforms on 1 OOD split but usually has advantages on ID tests. Finally, IRM performs similarly to ERM and outperforms on 3 of the OOD results, showing the difficulty of achieving invariant prediction in non-linear settings.

Graph OOD methods make extra effort to interpolate the irregularity and connectivity of graph topology, and certain improvements are achieved. Mixup-For-Graph exclusively excels at node prediction tasks, yielding consistent gains across datasets, which can attribute to its node-specific design [46]. It outperforms 6 out of 14 node-task OOD splits. However, it fails at graph prediction tasks due to the simple graph representation mixup strategy. DIR specifically solves concept shifts for graph classification tasks and outperforms on 3 splits, indicating that interventional augmentation on representations weakens spurious correlations by diversifying the distribution. Its benefit on concept shift does not apply to covariate shifts since DIR only expands the combination of representations without creating new domains; it also fails on regression tasks which require a more delicate learning process. EERM and SRGNN generally have average performances, outperforming only on a few splits. EERM reveals that while environment generation is learnable with REINFORCE, this adversarial training is difficult and needs to be perfected. SRGNN makes use of our OOD validation data to draw the training data closer to an OOD distribution; however, without sufficient generalization, it can seldom perform well in tests since OOD validation data cannot exactly reflect OOD test data. To conclude, while these graph-specific methods apply well to graph topology, other flaws in the methodology design create a performance bottleneck.

## 6 Discussions

GOOD aims to facilitate the development of graph OOD and general OOD algorithms. Our results and comparisons show that current OOD algorithms can improve generalization abilities, but not significantly. In addition, an algorithm might improve performance on one type of shift, but not both. With these observations, future OOD methods can focus on solving one of covariate and concept shifts to improve the specific generalization ability. The improvement might be achieved by managing well-designed model architectures, optimization schemes, or data augmentation strategies. Moreover, models cannot be expected to solve unknown distribution shifts. Thus, we believe using the given environment information to convey the types of shifts expected during testing is a promising direction.

Our GOOD benchmark is a growing project. We expect to include more methods as the OOD field develops especially graph-specific algorithms and include datasets and domain selections of a larger quantity and variety. In addition, the current benchmark does not consider link prediction tasks [58], which will be added as the project develops.

## Acknowledgments and Disclosure of Funding

We thank Jundong Li and Jing Ma for insightful discussions. This work was supported in part by National Science Foundation grants IIS-1955189, IIS-1908198, and IIS-1908220.

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
