# OpenReview forum: "GOOD: A Graph Out-of-Distribution Benchmark"
_NeurIPS.cc/2022/Track/Datasets_and_Benchmarks — NeurIPS 2022 Datasets and Benchmarks _

### Official Review · Reviewer_Ytr5 · 2022-07-11
**A potentially impactful work but reuqires more improvements**

**Rating:** 6
**Confidence:** 4

**Strengths:**

The authors systematically divide and discuss the distribution shifts into two categories, i.e., covariate shift and concept shift, to differentiate this work from other benchmarks for OOD generalization on graphs.

**Weaknesses:**

The main weakness of this work is the lack of graph specific discussions. In fact, throughout the paper, there are few specific designs and discussions about the unique challenges and distinctions of OOD generalization on graphs, which weakens the significance and novelty of the benchmark as it may not reflect the difficulties of OOD generalization on graphs. See more details below:
- The discussion about relevant works are limited. This paper has many overlaps with [1] and takes similar split strategies existing in the literature [1,2,3,4,5]. Why would the community prefer the proposed benchmark in this work instead of [1]? More discussions and comparisons are expected.
- The usage of causal languages in this work seems to be confusing. Given no causal graphs presented, how to determine the invariant features and variant features? Why the variant feature is causally irrelevant to the labels? If so, how would they correlate with the labels? The authors are suggested to refer [3,6] as well as the invariant learning literature for precise use of the causal languages.
- The definition of covariate shifts and concept shifts seem to be problematic too. For example, covariate shifts require P(Y|X) invariant across different environments/domains, but the introduction of X_inv and P(Y|X_inv) invariant, makes the P^tr(Y|X)=P^te(Y|X) confusing.
- In graphs, how to select invariant and variant features by index?
- This work seems to have weak relevance to graphs. The design of distribution shifts (Sec. 3) actually applies to both regular data and graph data, and has a poor reflection with the curation process of the datasets. For example, what kind of the scaffold, degree, size shifts belong to and why?
- This work does not benchmark any existing graph OOD methods, making the OOD gaps suspected. If the existing OOD methods on graphs [3,4,5,6] can solve the OOD shifts introduced in this benchmark already, developing follow-up works based on this benchmark seems to be uninteresting.

==== Post Rebuttal ====

I thank the authors for the detailed reply. Although they didn't fully resolve my concerns, I am convinced that this benchmark could be a good initial testbed for OOD algorithms on graphs. Thus I'd like to raise my score to Weak Accept.

My remaining concerns are still about the weak relatedness to graphs and the design of covariate and concept shifts as partially admitted by the authors, but I also admit that the novelty and contribution of the current design outweigh its weakness.

===================

References:

[1] Mucong Ding, Kezhi Kong, Jiuhai Chen, John Kirchenbauer, Micah Goldblum, David Wipf, Furong Huang, and Tom Goldstein. A closer look at distribution shifts and out-of-distribution generalization on graphs. In NeurIPS 2021 Workshop on Distribution Shifts: Connecting Methods and Applications, 2021.

[2] Yuanfeng Ji, Lu Zhang, Jiaxiang Wu, Bingzhe Wu, Long-Kai Huang, Tingyang Xu, Yu Rong, Lanqing Li, Jie Ren, Ding Xue, et al. DrugOOD: Out-of-distribution (OOD) dataset curator and benchmark for AI-aided drug discovery–a focus on affinity prediction problems with noise annotations. arXiv 2022.

[3] Beatrice Bevilacqua, Yangze Zhou, and Bruno Ribeiro. Size-invariant graph representations for graph classiﬁcation extrapolations. ICML 2021.

[4] Yingxin Wu, Xiang Wang, An Zhang, Xiangnan He, Tat-Seng Chua. Discovering Invariant Rationales for Graph Neural Networks. ICLR 2022.

[5] Qitian Wu, Hengrui Zhang, Junchi Yan, David Wipf. Handling Distribution Shifts on Graphs: An Invariance Perspective. ICLR 2022.

[6] Yongqiang Chen, Yonggang Zhang, Yatao Bian, Han Yang, Kaili Ma, Binghui Xie, Tongliang Liu, Bo Han, James Cheng. Invariance Principle Meets Out-of-Distribution Generalization on Graphs. arXiv 2022.



**Additional Feedback:**

Overall, I believe this work has the potential to be impactful, given the lack of a systematic benchmark of OOD generalization on graphs.
- However, the current version still requires more revisions to introduce more distinctions of OOD generalization on graphs compared to that on regular data.
- The languages to describe the distribution shifts seem to have conflicts with the literature on covariate shifts/concept shifts, and invariant learning.

The authors are suggested to refer [6] for an in-depth discussion of the two points above.


**Clarity:**

The paper is well-organized but hard to follow, given the confusing descriptions of the distribution shifts.

**Correctness:**

The languages used to describe distribution shifts on graphs are confusing. The experiments are not convincing given the lack of benchmarking existing OOD methods on graphs. See the Weakness for more details.

**Documentation:**

Yes.

**Relation To Prior Work:**

The discussion are limited. See the Weakness for more details.

**Summary And Contributions:**

This paper aims to develop an OOD benchmark for graphs. By dividing distribution shifts on graph into two types, the authors use existing datasets to artificially inject corresponding shifts. They evaluate existing OOD methods mostly from Euclidean data and demonstrate the IID-OOD generalization gap in the curated benchmark.

---

> ### Author Response · Authors · 2022-08-21
> **Major revision has been made based on your suggestions including the causal graph and graph specific discussions (Part I)**
>
> Dear reviewer Ytr5,
>
> Thank you for your comprehensive comments and constructive suggestions, which help to improve our work a lot! We have conducted substantial experiments and also revised the paper heavily. We also provide responses for each concern here.
>
>
> > 1. The main weakness of this work is the lack of graph specific discussions. In fact, throughout the paper, there are few specific designs and discussions about the unique challenges and distinctions of OOD generalization on graphs, which weakens the significance and novelty of the benchmark as it may not reflect the difficulties of OOD generalization on graphs.
>
> - Thank you for this constructive comment. We have revised the paper heavily to expand the graph-specific OOD discussion. Overall, considering graph-specific discussions, we have added the motivation and uniqueness of graph OOD in Section 1, the novelty of our benchmark in Section 2, specific shift splitting design of graphs in Section 3, experiments and analysis on graph-specific OOD methods in Section 5, and graph dataset processing details in Appendix A. We will also specify some points in the following parts of this response.
> - We included the unique challenges and distinctions of OOD generalization on graphs in Section 1 of the revised paper. Traditional OOD methods commonly focus on simple structure equation models or computer vision tasks. In these tasks, the inputs are variables or image features. However, graph data possess the complex nature of irregularity and connectivity in topology. A graph is commonly represented as an input pair $(F, A)$, where $F$ are node features, and $A$ is the adjacency matrix. Consequently, the graph OOD problem focuses not only on general feature distribution shifts but also on structure distribution shifts. Graph neural networks are designed for $F$ and $A$ to pass messages with different strategies, demonstrating that structure and feature comprehensively carry different perspectives of input information of a graph. The uniqueness of graph data prompts the development of graph-specific OOD methods and calls for graph OOD benchmarks.
> - To well-reflect the difficulties of OOD generalization on graphs, we have added and evaluated three more graph-specific OOD methods, DIR[1], EERM[2], SR-GNN[3], which give GOOD the sufficiency to reflect the performances and capabilities of current graph-specific OOD methods. DIR is an OOD algorithm for graph-classification tasks, while EERM and SRGNN are designed for node-classification tasks. With extra experimental results on these graph-specific OOD methods, we conclude that while these graph-specific methods apply well on graph topology, other flaws in the methodology design create a performance bottleneck, limiting the OOD performance outcome. Therefore, the performance gap between ID and OOD remains, and better graph OOD methods are expected, evidencing our benchmark's significance.

---

> > ### Author Response · Authors · 2022-08-21
> > **Major revision has been made based on your suggestions including the causal graph and graph specific discussions (Part II)**
> >
> > > 2. The discussion about relevant works are limited. This paper has many overlaps with GDS and takes similar split strategies existing in the literature. Why would the community prefer the proposed benchmark in this work instead of GDS? More discussions and comparisons are expected.
> >
> > - Many thanks for this very useful suggestion. To clarify this concern, we have added substantial discussions and comparisons in the Related Work Section. Please refer to the revised paper for details. We also conclude the comparison as follows.
> > - Generalization abilities of OOD algorithms for covariate and concept shifts differ fundamentally. However, existing benchmarks, including but not limited to graph OOD benchmarks, either ignore one of the shifts or fail to compare the two shifts of the same feature **on the same dataset**. To the best of our knowledge, **concept shift splitting process** is first proposed by our paper. With the help of the new splitting process, we are the first benchmark to compare covariate shift and concept shift **on the same domain**, making the shift consideration **complete**. Most existing OOD benchmarks include only one type of shift. For example, DrugOOD[14] exclusively considers covariate shift. WILDS[15] includes two cases of covariate shift, while still ignoring concept shift. A few benchmarks involve both shifts, but each dataset has **only one of these two shifts**. GDS[13] collects eight datasets but makes no distinctions between them and does not even mention the two shifts. In contrast, our benchmark proposes novel dataset splitting methods to generate **both shifts for the same domain selection** on the same dataset to enable comparison between shifts. To the best of our knowledge, no existing OOD benchmark provides such a comparison.
> > - Compared to existing graph OOD benchmarks, except for all the advantages explained above, we are also far more diverse in tasks. Neither GDS[13] nor DrugOOD[14] includes any graph regression or node classification tasks, while we curate more diverse graph datasets with diverse tasks, including single/multi-task graph classification, graph regression, and node classification. In addition, while GDS and DrugOOD do not benchmark any graph-specific OOD methods, we evaluate 4 graph-specific OOD methods, shedding light on further graph OOD research.
> >
> >
> > > 3. The usage of causal languages in this work seems to be confusing. Given no causal graphs presented, how to determine the invariant features and variant features? Why the variant feature is causally irrelevant to the labels? If so, how would they correlate with the labels? The authors are suggested to refer [3,6] as well as the invariant learning literature for precise use of the causal languages.
> >
> > Thank you for your suggestion! Initially, we tried to make this part readable for more general users, but it caused much confusion, as you mentioned. Therefore, we added a causal graph in the revision to make descriptions clearer. We believe the added causal graph can address these questions.
> >
> > In our revised paper, we divide input features into three parts instead of two parts: invariant part, associated part, and independent part. And the domain selection for covariate shift will be the independent part. Since $Y\bot X_{ind}$ the change of $P(X_{ind})$ will not cause the change of $P(Y|X)$. As we mentioned in the revised paper, it is not necessary or possible to distinguish them clearly in practice. We use these notations mainly for theoretical analysis. Similar to prior OOD benchmarks, human intuition will be the major practical strategy for domain selection.
> >
> > > 4. The definition of covariate shifts and concept shifts seem to be problematic too. For example, covariate shifts require P(Y|X) invariant across different environments/domains, but the introduction of X_inv and P(Y|X_inv) invariant, makes the P^tr(Y|X)=P^te(Y|X) confusing.
> >
> > - Yes, we realized that the introduction of X_inv and P(Y|X_inv) invariant is confusing. We have modified it in the revised paper.
> >
> >
> > > 5. In graphs, how to select invariant and variant features by index?
> >
> > - As described in our revised introduction and footnote 2, since a formatted graph input includes node features and adjacency matrix, we consider higher-level features that can represent structure information. As shown in Figure 2 of the revision, $X$ represents the features that humans can partially observe, including motif shapes or molecule scaffolds. Therefore, in practice, we do not select features by index; instead, we use algorithms to extract the feature we need. For instance, if we need the scaffold feature of a molecule, we will apply the Bemis-Murcko scaffold algorithm [16]. To eliminate possible confusion, we avoid introducing the feature index.

---

> > > ### Author Response · Authors · 2022-08-21
> > > **Major revision has been made based on your suggestions including the causal graph and graph specific discussions (Part III)**
> > >
> > > > 6. This work seems to have weak relevance to graphs. The design of distribution shifts (Sec. 3) actually applies to both regular data and graph data, and has a poor reflection with the curation process of the datasets. For example, what kind of the scaffold, degree, size shifts belong to and why?
> > >
> > > Thank you for your suggestions!
> > >
> > > - To eliminate the graph relevance concern, we added three carefully selected graph-specific OOD algorithms in our revised paper. Therefore, we included more thorough discussions about the graph OOD problem. Moreover, we hope to explain that one of the keys to the curation process, the domain selection process, is highly correlated with graphs. As the last question explains, the graph "feature" is calculated by applying graph-specific algorithms instead of using a formatted input feature directly. Therefore, the curated datasets can reflect the graph's exclusive topological shift.
> > >
> > > - The design of distribution shifts does apply to both regular data and graph data, but we would like to respectfully clarify that this design is also where our novelty locate. To the best of our knowledge, **concept shift splitting process** is first proposed by our paper. With the help of a new splitting process, we are the first benchmark that can compare covariate shift and concept shift **on the same domain**. We also claim that since $P(Y,X)=P(Y|X)P(X)$, the shift consideration of covariate and concept shifts is **complete**, enabling **thorough comparisons of possible shifts on every given domain**. These contributions are original not only in the field of graph OOD but also in the field of general OOD.
> > > - To have a better reflection on the curation process of the datasets, we added the discussion "Comparison of the covariate(/concept) split design on 3 types of datasets" at the end of Sections 3.1 and 3.2, respectively, which we explain in detail the shift processing on datasets. Please refer to the revised paper for details.
> > > - For each of our domain selections (e.g., scaffold, size, color, time, degree, and so on,) we perform covariate shift split and concept shift split **on the same domain**. This is enabled by our method of shift split design. Therefore, for each selected domain, e.g., scaffold, we can observe and compare the performances of each algorithm on "scaffold-covariate" and "scaffold-concept". The practical meaning and split details of graph datasets are included in Section 4 and Appendix A, and the graph-specific analysis of both shifts is included in Section 5.2.

---

> > > > ### Author Response · Authors · 2022-08-21
> > > > **Major revision has been made based on your suggestions including the causal graph and graph specific discussions (Part IV)**
> > > >
> > > > > 7. This work does not benchmark any existing graph OOD methods, making the OOD gaps suspected. If the existing OOD methods on graphs [3,4,5,6] can solve the OOD shifts introduced in this benchmark already, developing follow-up works based on this benchmark seems to be uninteresting.
> > > >
> > > > - To address this concern, and also coincided with the updated version of GOOD we've been working on, we have added **three other graph-specific OOD algorithms,
> > > > DIR[1], EERM[2], SR-GNN[3]**, which give GOOD the sufficiency to reflect the performances and capabilities of current graph-specific OOD methods. DIR is an OOD algorithm for graph-classification tasks, while EERM and SRGNN are designed for node-classification tasks. **It is noteworthy that the Mixup method in our paper is a graph-specific Mixup [17].**
> > > > - In addition, with extra experimental results on these graph-specific OOD methods, we conclude that while these graph-specific methods apply well to graph topology, other flaws in the methodology design create performance bottleneck, limiting the OOD performance outcome. Therefore, the performance gap between ID and OOD remains, and better graph OOD methods are expected.
> > > > - We provide justifications for our selections. We select those three algorithms for 3 reasons.
> > > >  - These algorithms have passed peer review and been accepted by major conferences or journals.
> > > >  - The code of these algorithms is well-maintained; thus, they are reproducible.
> > > >  - These algorithms target solving the OOD problem on graphs.
> > > >
> > > >  Among all other graph OOD methods within our knowledge, graph OOD algorithms like [4, 5, 6, 7, 8] haven't been accepted by major conferences or journals yet, while the disentangled GNN methods [9, 10, 11] do not target solving the OOD problem. While [12] is an interested and solid paper we would like to implement, the key code for obtaining induced homomorphism densities is missing. The authors claimed to use R-GPM in their paper, but some C++ compile files are missing in R-GPM's GitHub repository, and the API used in the code mismatches with R-GPM's. The missing and mismatching disabled the reproducibility of this algorithm. In conclusion, we believe our selection of graph-specific algorithms is sufficient and of quality.
> > > > - The introduction of algorithms, performance comparisons, and algorithm supplements of the algorithms are updated in Section 5.2, Table 2, and supplementary material of the revised paper. The details are also elaborated in the first improvement of the [Common Response](https://openreview.net/forum?id=8hHg-zs_p-h&noteId=fszCjK8zqx) above.
> > > >
> > > >
> > > >
> > > > We sincerely thank you for your time! Hope we have addressed your concerns through practical efforts. We look forward to your reply and further discussions, thanks!
> > > >
> > > > Sincerely,
> > > >
> > > > Authors

---

> > > > > ### Author Response · Authors · 2022-08-21
> > > > > **References**
> > > > >
> > > > > [1] Wu, Y.-X., Wang, X., Zhang, A., He, X., & Chua, T.-S. (2022). DISCOVERING INVARIANT RATIONALES FOR GRAPH NEURAL NETWORKS. ICLR 2022.
> > > > >
> > > > > [2] Wu, Q., Zhang, H., Yan, J., & Wipf, D. (2022). HANDLING DISTRIBUTION SHIFTS ON GRAPHS: AN INVARIANCE PERSPECTIVE. ICLR 2022.
> > > > >
> > > > > [3] Zhu, Qi, et al. "Shift-robust gnns: Overcoming the limitations of localized graph training data." Advances in Neural Information Processing Systems 34 (2021): 27965-27977.
> > > > >
> > > > > [4] Sui, Y., Wang, X., Wu, J., He, X., & Chua, T.-S. (2021). Deconfounded Training for Graph Neural Networks. Arxiv. http://arxiv.org/abs/2112.15089
> > > > >
> > > > > [5] Wang, Xiang, et al. "Deconfounding to Explanation Evaluation in Graph Neural Networks." arXiv preprint arXiv:2201.08802 (2022).
> > > > >
> > > > > [6] Fan, Shaohua, et al. "Generalizing Graph Neural Networks on Out-Of-Distribution Graphs." arXiv preprint arXiv:2111.10657 (2021).
> > > > >
> > > > > [7] Zhang, Shengyu, et al. "Stable Prediction on Graphs with Agnostic Distribution Shift." arXiv preprint arXiv:2110.03865 (2021).
> > > > >
> > > > > [8] Chen, Yongqiang, et al. "Invariance Principle Meets Out-of-Distribution Generalization on Graphs." arXiv preprint arXiv:2202.05441 (2022).
> > > > >
> > > > > [9] Yang, Yiding, et al. "Factorizable graph convolutional networks." Advances in Neural Information Processing Systems 33 (2020): 20286-20296.
> > > > >
> > > > > [10] Ma, Jianxin, et al. "Disentangled graph convolutional networks." International conference on machine learning. PMLR, 2019.
> > > > >
> > > > > [11] Li, Haoyang, et al. "Disentangled contrastive learning on graphs." Advances in Neural Information Processing Systems 34 (2021): 21872-21884.
> > > > >
> > > > > [12] Bevilacqua, Beatrice, Yangze Zhou, and Bruno Ribeiro. "Size-invariant graph representations for graph classification extrapolations." International Conference on Machine Learning. PMLR, 2021.
> > > > >
> > > > > [13] Mucong Ding, Kezhi Kong, Jiuhai Chen, John Kirchenbauer, Micah Goldblum, David Wipf, Furong Huang, and Tom Goldstein. A closer look at distribution shifts and out-of-distribution generalization on graphs. In NeurIPS 2021 Workshop on Distribution Shifts: Connecting Methods and Applications, 2021.
> > > > >
> > > > > [14] Yuanfeng Ji, Lu Zhang, Jiaxiang Wu, Bingzhe Wu, Long-Kai Huang, Tingyang Xu, Yu Rong, Lanqing Li, Jie Ren, Ding Xue, et al. DrugOOD: Out-of-distribution (OOD) dataset curator and benchmark for AI-aided drug discovery–a focus on affinity prediction problems with noise annotations. arXiv 2022.
> > > > >
> > > > > [15] Pang Wei Koh, Shiori Sagawa, Henrik Marklund, Sang Michael Xie, Marvin Zhang, Akshay Balsubramani, Weihua Hu, Michihiro Yasunaga, Richard Lanas Phillips, Irena Gao, et al. Wilds: A benchmark of in-the-wild distribution shifts. In International Conference on Machine Learning, pages 5637–5664. PMLR, 2021
> > > > >
> > > > > [16] Bemis, Guy W., and Mark A. Murcko. "The properties of known drugs. 1. Molecular frameworks." Journal of medicinal chemistry 39.15 (1996): 2887-2893.
> > > > >
> > > > > [17] Wang, Yiwei, et al. "Mixup for node and graph classification." Proceedings of the Web Conference 2021. 2021.

---

> ### Author Response · Authors · 2022-08-25
> **Author's follow up to reviewer Ytr5**
>
> Dear Reviewer Ytr5,
>
> Thanks again for your valuable comments and suggestions in your initial review, which helps improve our work a lot. Regarding your main concerns on the lack of graph specific discussions and the confusion on causal descriptions and definitions, which we believe is extremely valuable and critical, we have conducted substantial experiments and also revised the paper heavily. We have also clarified all concerns sentence by sentence in our response. Could you please check at your earliest convenience? Thanks!
>
> We hope that you could reply to our response and revision and consider raising your score if we do have addressed your concerns. Also, please let us know if there are any additional concerns or feedback. Thank you very much!
>
> Sincerely,
>
> Authors

---

> > ### Comment · Reviewer_Ytr5 · 2022-08-27
> > **Thanks authors for the tremendous rebuttal efforts**
> >
> > Dear authors,
> >
> > Thanks for your tremendous efforts during the rebuttal phase. Apologize for the late reply as it took much time to go through all of the revisions and the replies. Nevertheless, I still have some concerns.
> >
> > 1. About the relatedness of the graphs:
> > Although making up some of the graph OOD algorithms in benchmarking resolve part of my concerns, the current explanation about the unique challenges of OOD generalization on graphs seems to be weakly related to the design of the benchmarks. More specifically,
> >
> > - How, and to what extent, does the design of the benchmark reflect the challenges of the graph OOD? It seems the graph OOD is not more difficult than general OOD, as, in most cases, general OOD algorithms stand out as state-of-the-art.
> > - How are the challenges in graph OOD related to the design of concept and covariate shifts?
> > - How much do the designed distribution shifts reflect the *real-world* graph OOD? In practical applications, do they really exist?
> >
> > 2. About the description and curation of the distribution shifts:
> > - Can the authors formally define all of the variables used in the paper?
> > - Does the causal graph only apply to Covariate shifts, as the authors put its description in Sec. 3.1? The causal graph seems not to follow the standard causal languages, could you improve it?
> > - Can distribution shifts also happen to $X_\text{ass}$? If so, how can you claim that "making the shift consideration complete." (in the reply)? Considering distribution shifts caused by $X_\text{ass}$: In general, $X_\text{ass}$ controlled by $C$ and $Y$ can exhibit substantially different behaviors and need to be carefully differentiated, as discussed in IB-IRM (Ahuja et al., 2021) cited in the paper.
> > - How can the curation guarantee that the selected features are $X_\text{ind}$ as described in the causal graph? Couldn't some of them belong to $X_\text{ass}$?
> > - In Sec. 4.1, the authors seem to make strong assumptions on the selected "$X_\text{ind}$" for each dataset. Here are few examples,
> >     - For molecule datasets, "Both features should not determine the label, therefore, both can become major sources of distribution shifts.". To my best understanding, "Shouldn't determine" can only exclude $X_\text{inv}$.
> >     - For GOODSST2, "We select sentence lengths as domains since the length of a sentence should not affect the sentimental polarity", could the sentence lengths belong to $X_\text{ass}$?
> >     - For GOOD-Cora, "word diversity defined by the selected-wordcount of a publication, purely irrelevant with the label " seems to be carefully verified.
> >    - For GOOD-Twitch, "The domain of GOOD-Twitch splits includes user language, implying that the prediction target should not be biased by the language a user uses." What if users that use certain languages would have certain preferences?
> >
> > In fact, careful discussions and verifications on how the generative process of the corresponding datasets correspond to the causal models established in the paper seem to be needed, and how these distribution shifts can reflect challenges in graph OOD, seem to be needed.
> >
> > 3. Other technical questions:
> > - In Table 21-31, it seems the hyperparameter search spaces are a bit limited (smaller than many of the existing OOD datasets and those provided by the original papers). How can the benchmark ensure the fairness of comparison?
> > - How can one determine the overall OOD generalization ability of different algorithms?
> > - What are the exact values of domain selection features, e.g., size ranges? I couldn't find one in the paper.
> >
> > To summarize, I do appreciate the huge amount of work of the authors to curate the benchmark, and this work can have a high potential to impact and guide future developments in this field. But I feel it requires substantial revisions to clarify the motivation of the current design and the comparison protocol of the benchmark. Thus I still lean to reject the current version of the paper.

---

> > > ### Author Response · Authors · 2022-08-28
> > > **Thank you for your reply. We would like to clarify your concerns and discuss why this work should be accepted. (Part I)**
> > >
> > > Dear reviewer Ytr5,
> > >
> > > Thank you for your enormous efforts and earnest replies! We believe that we are on the same page and share some same concerns. We are glad to have an in-depth discussion with you. We would like to clarify and eliminate your concerns point by point respectfully.
> > >
> > > ### 1. About the relatedness of the graphs: Although making up some of the graph OOD algorithms in benchmarking resolve part of my concerns, the current explanation about the unique challenges of OOD generalization on graphs seems to be weakly related to the design of the benchmarks.
> > >
> > > > * How, and to what extent, does the design of the benchmark reflect the challenges of the graph OOD? It seems the graph OOD is not more difficult than general OOD, as, in most cases, general OOD algorithms stand out as state-of-the-art.
> > >
> > > Firstly, as we mentioned in our revised introduction, the graph OOD is more difficult than the general OOD mainly because of the complexity in the shift of structure. The design of our benchmark reflects this point well since many domain selections are structure-related. Secondly, it is true that performances show general OOD algorithms stand out as state-of-the-art in many cases. **However, it cannot justify that "the graph OOD is not more difficult than general OOD". Instead, the reason that general OOD algorithms can outperform graph-specific algorithms is the design defects of graph-specific algorithms,** though they fit graph topology. For example, DIR aims to use masks to choose a causal subgraph and makes the prediction. The results show that the mask is difficult to learn, and a wrongly learned mask (subgraph) will contribute to worse performance. Therefore, it further shows the difficulty of tackling structure-related shifts, which belong to the graph OOD problem.
> > >
> > > > * How are the challenges in graph OOD related to the design of concept and covariate shifts?
> > >
> > > Our novel design of concept and covariate shifts can apply to all OOD problems, and we believe that our contributions benefit not only the graph OOD but also the general OOD. This design is not limited to graphs.
> > > The graph OOD challenges and difficulties are as aforementioned, and we focus on graph OOD to benefit graph learning. Therefore, they are not originally related. They are both discussed in our paper largely because we want to make a more comprehensive OOD benchmark and we focus on graphs.
> > >
> > > > * How much do the designed distribution shifts reflect the real-world graph OOD? In practical applications, do they really exist?
> > >
> > > We consider it critical that the designed splits reflect real-world scenarios; therefore, all our domain selections are based on commonly addressed OOD problems in practice. For example, molecules with some scaffolds can be easily obtained, but molecules with other scaffolds are hard to be found because of the limitation of the experiment environments. Therefore, we choose scaffolds as domains and simulate this situation. For another example, we can simulate a situation where people with negative sentiments are required to give longer reasonable comments, while positive people are not required to do that. In another environment, positive comments are required to include detailed reasons, while negative ones do not. Therefore, there is a concept shift between the sentiment polarity and the comment's length. Overall, we believe our designed splits well reflect the real-world graph OOD.
> > >
> > > In the real world, the covariate shift and concept shift may appear in combination, and multiple shifts of different domains may appear together. We make variable control to our best, controlling single domain and shift type with the proper splitting process. This enables the evaluation of a method with a specific domain and shift type, which benefits the development of graph OOD methods since most methods show different generalization abilities on different shifts.

---

> > > > ### Author Response · Authors · 2022-08-28
> > > > **Thank you for your reply. We would like to clarify your concerns and discuss why this work should be accepted. (Part II)**
> > > >
> > > > ### 2. About the description and curation of the distribution shifts:
> > > >
> > > > > Can the authors formally define all of the variables used in the paper?
> > > >
> > > > Yes, we have defined them more formally in the revised paper, and we list the major part as follows. Please refer to the revised paper for more details.
> > > >
> > > > "$C$ is the latent variable that causes target $Y$; $S_1$ is a latent variable non-causally associated with $Y$. Additionally, we consider a $S_2$ independent of $Y$, which is a hidden variable commonly caused by target-irrelevant environments. In the input feature layer, given input features $X$, we assume the invariant features $X_\text{inv}\in X$ are projected by an injective function from $C$. $X_\text{ass}$ denotes input features associated with $Y$ by confounding and anti-causal associations through $C$ and $S_1$. $X_\text{ind}$ are input features independent to $Y$ and are only caused by $S_2$".
> > > >
> > > > > Does the causal graph only apply to Covariate shifts, as the authors put its description in Sec. 3.1?
> > > >
> > > > No, we would like to clarify that the causal graph applies to both covariate shift and concept shift, and we continue using the same notations in the concept shift section.
> > > >
> > > > > The causal graph seems not to follow the standard causal languages, could you improve it?
> > > >
> > > > Yes, we have modified the description of the causal graph to improve the readability. We follow Pearl's causal language [2] to make descriptions. Please let us know if there are further concerns. For discussion efficiency, we would like to respectfully provide possible explanations of your concerns:
> > > >
> > > > * Our causal graph not only considers those hidden variables as other works, but also needs to consider the quasi-input representation layer: $X_{inv}$, $X_{ass}$, and $X_{ind}$. The reason why our causal graph is different with [1, 8] is that these two works does not distinguish the $C$, $S$ and $X_{inv}, X_{ass}, X_{ind}$, since they don't need to discuss $X_{inv}, X_{ass}, X_{ind}$. Nevertheless, as a benchmark work, since we apply dataset split (secondary dataset processing), we need to distinguish them further. Specifically, $C, S, Y$ part happens in the real world, and we cannot reproduce their causations after **observation**. When we apply the dataset split, what we can get are only parts of $X_{inv}, X_{ass}, X_{ind}$. Considering the significant difference compared to [1, 3]: when they claim that they try to capture the $C$ and $S$ reversely, it corresponds to the practical dataset split process as $X_{inv}, X_{ass}, X_{ind}$.
> > > > * As we know, even though some works use causality literature, causality in deep learning is still changing over time, unlike other mature works like Bayesian network. The development of causal graphs for deep learning is still in progress. As the case in our work, we need to distinguish the $C,S,Y$ part and the $X_{inv}, X_{ass}, X_{ind}$ part. Also, we need to assume that $C→X_{inv}$ is an injective function so that the input has the potential to predict $Y$. Furthermore, we believe that our descriptions follow Pearl's causal language. For example, the explanation of the *do-operation* in the Q&A of Pearl's book [2] is much more flexible than that when it is first introduced.
> > > >
> > > > > Can distribution shifts also happen to $X_{ass}$? If so, how can you claim that "making the shift consideration complete." (in the reply)? Considering distribution shifts caused by $X_{ass}$: In general, $X_{ass}$ controlled by C and Y can exhibit substantially different behaviors and need to be carefully differentiated, as discussed in IB-IRM (Ahuja et al., 2021) cited in the paper.
> > > >
> > > > The claim of "complete" is under the **context of** $P(X,Y)=P(X)P(Y|X)$ where we consider both $P(X)$ and $P(Y|X)$. From this point, with the covariate and concept shift, our benchmark is complete. However, the OOD problem is complicated and diverse; with **finer-grained consideration**, one may think that though GOOD is comprehensive, it can still cover more specific cases, which is not our perspective of discussing **completeness**. Therefore, we believe that our claim is valid under the context we discuss.
> > > >
> > > > We do not consider distribution shifts on $X_{ass}$ theoretically.
> > > > If we consider $X_{ass}$, there can be no pure covariate shift, since $X_{ass}$ would involve $Y$. Therefore, we only consider covariate shift on $X_{ind}$. Further, since we propose to compare covariate and concept shifts **on the same domain**, both shifts only consider the cases with $X_{ind}$.
> > > > In addition, intuitively, selecting features that are $X_{ind}$ is much more practical than selecting $X_{ass}$. The justification that the selected features are $X_{ind}$ is detailed in the following question.

---

> > > > > ### Author Response · Authors · 2022-08-28
> > > > > **Thank you for your reply. We would like to clarify your concerns and discuss why this work should be accepted. (Part III)**
> > > > >
> > > > > > How can the curation guarantee that the selected features are $X_{ind}$ as described in the causal graph? Couldn't some of them belong to $X_{ass}$? In Sec. 4.1, the authors seem to make strong assumptions on the selected "$X_{ind}$" for each dataset.
> > > > >
> > > > > Thank you for your comments. We want to share two considerations to eliminate your concern respectfully:
> > > > >
> > > > > * **The answers to these questions are that the domain selection is manually picked. It depends on human intelligence.** We believe that there is no better solution with current knowledge to guarantee that the selected features are $X_{ind}$. One may argue that we can use the empirical independence measure to test whether $Y$ is independent of $X_{ind}$. However, how can one justify that the original datum is sampled without any shift or biases? No measure can verify whether $X_{ind}$ is causally irrelevant with $Y$ in real-world scenarios, which is pretty much the same case as manual dataset annotation. In some complicated dataset annotation tasks, what we can state is that the annotations come from a human.
> > > > > We assumption that the selected features are $X_{ind}$, because if we consider $X_{ass}$, the covariate shift will not hold since $X_{ass}$ would involve $Y$.
> > > > > - These questions actually were our concern before, so we are glad you provide this in-depth comment. However, we gradually realize that this tricky question will lead us to a dead end.
> > > > > To push the case to a more extreme situation, for example, though **intuitively "scaffold should not determine the label"**, we can even doubt **whether scaffold contain a part of** $X_{inv}$, since containing only a part of $X_{inv}$ would still not "determine the label".
> > > > > Other than empirical experiments, it is not even mathematically provable that the scaffold of a molecule does not contain a part of $X_{inv}$. However, if the domain feature includes parts of $X_{inv}$, the data in the test set may lack invariant features, leading to the lack of potential to make predictions. The consequence is that all current OOD benchmarks are unreliable except for random splits. Therefore, to practically obtain meaningful data, we believe there is no better solution with current knowledge to guarantee that the selected features are $X_{ind}$. We manually make selections to reflect real-world scenarios with each dataset, for example, in GOOD-Twitch, we aim at achieving mature content detection with **fairness**, free from languages bias.
> > > > >
> > > > > * **The good aspect is that, in practice, the selection of $X_{ind}$ needs not to be so strict as the theoretical analysis.** Even if the picked feature is associated with $Y$, as long as this association is weak, the dataset splits will largely have a covariate shift or concept shift. **This is because our dataset split process will build strong shifts which can dominate over other weak shifts. Then these splits can still benchmark the covariate shift and concept shift performances of algorithms.** Still, we endeavor to select domain features that have as lower associations with $Y$ as possible.
> > > > >
> > > > > To be more convincing, we can take the GOOD-SST2 and GOOD-Twitch as examples. In GOOD-SST2, we can simulate both environments where positive comments are dominating, or negative comments are dominating. In GOOD-Twitch, we can build extreme preferences according to language. These additional shifts are much stronger than the original shift, and we can clearly see the algorithm's different behaviors across different shifts.
> > > > >
> > > > > As a benchmark, we believe that our purpose for benchmarking algorithms for different shift preferences is well achieved.
> > > > >
> > > > > ### 3. Other technical questions:
> > > > >
> > > > > > In Table 21-31, it seems the hyperparameter search spaces are a bit limited (smaller than many of the existing OOD datasets and those provided by the original papers). How can the benchmark ensure the fairness of comparison?
> > > > >
> > > > > Thank you for your concern. We would like to clarify that the hyperparameters shown in the table have passed a round of filtering. As you may notice, the values of the hyperparameter finetune set vary across splits. This is not from random choice but the consequence of a wide range of prior hyperparameter attempts from 1e-10 to 10000.
> > > > > Many unreasonable hyperparameters are filtered out with early stop, and we keep the three values covering the best-performing range. All the hyperparameters in those tables are well selected, and they can only be picked out after full comparisons. Note that we tried not only all the hyperparameter selections of the original paper but also more than their selections. At last, we chose the most reasonable values and made the final selection. The fairness of our experiments can also be justified by the fact that we mostly present comparable higher results than other works with similar datasets and settings.

---

> > > > > > ### Author Response · Authors · 2022-08-28
> > > > > > **Thank you for your reply. We would like to clarify your concerns and discuss why this work should be accepted. (Part IV)**
> > > > > >
> > > > > > > How can one determine the overall OOD generalization ability of different algorithms?
> > > > > >
> > > > > > The OOD generalization is diverse. Currently, we have not noticed an algorithm that can tackle all these shifts. In the nature of their design, algorithms solve different aspects of OOD problems. Therefore, instead of drawing a "convenient" conclusion or providing an ad-hoc overall OOD generalization ability metric, we would like to encourage the development of diverse OOD perspectives.
> > > > > >
> > > > > > > What are the exact values of domain selection features, e.g., size ranges? I couldn't find one in the paper.
> > > > > >
> > > > > > We did not provide the exact values of domain selection features because many domain selection features cannot be represented as "values," and there are too many to be listed explicitly, *e.g.*, GOOD-HIV has 19,089 scaffolds, which is not practical to be all drawn out. To present all domains with consistency, we only provide "# domains" (Number of domains) in Fig.1 of the paper.
> > > > > > To provide an example of size ranges for your concern, for GOOD-HIV's size domain and covariate shift, the training set includes graphs with a size range of 15~150, averaging 25.84.
> > > > > >
> > > > > > Following the requirement of the dataset and benchmark track, all dataset details can be found at https://drive.google.com/drive/u/1/folders/1EcSGRkNxBLOUoRoLuhaazQZTKRGwaquX. They can be loaded by our detailed documented code in https://github.com/divelab/GOOD/.
> > > > > >
> > > > > > Thanks again for your comments! We really appreciate your detailed replies. You are absolutely a responsible reviewer! We hope we have addressed your major concerns. In addition, we would still like to discuss why our benchmark should be accepted briefly:
> > > > > >
> > > > > > * Our benchmark consists of sufficient contributions and can significantly benefit the graph OOD community. (Detailed novelties and motivations are described before.) Our contributions are also admitted by the reviewer R994 (rating from 5 to 6) and the reviewer SeXB (rating from 4 to 6).
> > > > > > * We believe that we have well clarified your 1st and 3rd concerns and sufficiently addressed the 2nd.
> > > > > > * The most controversial concern (the 2nd concern) is a trade-off between theory and practice. Since our benchmark can still achieve our purpose ( benchmarking algorithms for different shift preferences), we consider it not dominating over our contributions.
> > > > > >
> > > > > > We hope that we have well addressed your concerns and you could consider raising your score. Also, please let us know if there are any additional concerns or feedback. Thank you very much!
> > > > > >
> > > > > > Sincere,
> > > > > >
> > > > > > Authors
> > > > > >
> > > > > >
> > > > > > [1] Wu, Y.-X., Wang, X., Zhang, A., He, X., & Chua, T.-S. (2022). DISCOVERING INVARIANT RATIONALES FOR GRAPH NEURAL NETWORKS. ICLR 2022.
> > > > > >
> > > > > > [2] Pearl, Judea. Causality. Cambridge university press, 2009.
> > > > > >
> > > > > > [3] Chen, Yongqiang, et al. "Invariance Principle Meets Out-of-Distribution Generalization on Graphs." arXiv preprint arXiv:2202.05441 (2022).
> > > > > >
> > > > > > [4] Peters, Jonas, Dominik Janzing, and Bernhard Schölkopf. Elements of causal inference: foundations and learning algorithms. The MIT Press, 2017.

---

> > > > ### Comment · Reviewer_Ytr5 · 2022-08-28
> > > > **Thanks authors for the follow-up reply**
> > > >
> > > > I thank the authors for the detailed reply. Although they didn't fully resolve my concerns, I am convinced that this benchmark could be a good initial testbed for OOD algorithms on graphs. Thus I'd like to raise my score to Weak Accept.
> > > >
> > > > My remaining concerns are still about the weak relatedness to graphs and the design of covariate and concept shifts as partially admitted by the authors, but I also admit that the novelty and contribution of the current design outweigh its weakness.
> > > >
> > > > Nevertheless, I feel the authors could further improve many aspects of the current version:
> > > > - The authors could improve the readability of Sec. 3, by
> > > >     - improving the introduction of the causal graph with more formal languages (cf. [3,6] in my original reference and OoD-Bench by Ye et al., 2021), e.g., differentiating the observed and latent variables, defining variable spaces and etc. It can make the follow-up works better aware of the sources of the distribution shifts.
> > > >     - moving the introduction of the causal graph before the introductions of the specific distribution shifts seems to be clearer to readers.
> > > >     - aligning with the invariant learning literature better by introducing the relatedness of the Environment variable in previous literature and the current design of the distribution shifts.
> > > >
> > > > - It seems to be meaningful to include the discussions of the limitations regarding the selection of $X_{ind}$ in the paper. These philosophical discussions could shed some light on future developments of better benchmarks and algorithms.
> > > >
> > > > - It could facilitate the follow-up developments based on the benchmark if the authors could make the evaluation protocol clearer, e.g., hyperparameter selection, and include a clearer roadmap of the developments. It seems the authors haven't included the state-of-the-arts in graph OOD, e.g., [6] in my original reference and GSAT by Miao et al., 2022 (arxiv 2201.12987, and recently accepted by ICML 2022). Both these two works show that they can fully resolve the distribution shifts discussed in DIR paper. The current results show that DIR fails to achieve effective OOD, which seems to be explained by [6] that DIR failed to establish valid theoretical guarantees for the distribution shifts discussed in DIR paper.

---

> > > > > ### Author Response · Authors · 2022-08-29
> > > > > **Thank you for your acknowledgement. We would like to address the rest of your concerns with more clarifications and the further revision.**
> > > > >
> > > > > Dear Reviewer Ytr5,
> > > > >
> > > > > Many thanks to your detailed explanations and suggestions! The discussions improve our work a lot. We are glad to know that you believe the novelty and contribution of the current design outweigh its weakness.
> > > > > With your acknowledgment, we address your concerns further as follows.
> > > > >
> > > > > > My remaining concerns are still about the weak relatedness to graphs and the design of covariate and concept shifts.
> > > > >
> > > > > Even though we can also apply the current design of covariate and concept shifts for image datasets, similarly, the design of covariate and concept shifts also has weak relatedness to images. We believe the relatedness between graphs and the design of covariate and concept shifts is not dominatingly important in our work.
> > > > > These two shifts are a theoretical analysis to obtain completeness of distribution shifts, an improvement not specifically relating to a data form.
> > > > > Therefore, the design of covariate and concept shifts is serving for better graph OOD benchmarking, which is the priority of our work. We hope this explanation can be satisfying.
> > > > >
> > > > > > The authors could improve the readability of Sec. 3, by…
> > > > >
> > > > > Thanks a lot for your constructive and detailed suggestions. We revised our paper by 1) further formalizing the causal languages with definitions of causal relations, variable spaces, and the distinction of observable/unobservable variables. 2) We move the introduction of the causal graph right before introducing specific distribution shifts. 3) We add the environment variable into the causal graph that is aligned with our environment introduction. Please refer to the revised paper.
> > > > >
> > > > > > It seems to be meaningful to include the discussions of the limitations regarding the selection of $X_{ind}$ in the paper. These philosophical discussions could shed some light on future developments of better benchmarks and algorithms.
> > > > >
> > > > > Thanks. We have added discussions about the imperfect selection of $X_{ind}$ in revision. Please refer to the revised paper.
> > > > > <!-- But due to the space limit, we cannot extend this discussion too much in the main paper. -->
> > > > >
> > > > > > It could facilitate the follow-up developments based on the benchmark if the authors could make the evaluation protocol clearer, e.g., hyperparameter selection, and include a clearer roadmap of the developments. It seems the authors haven't included the state-of-the-arts in graph OOD, e.g., [6] in my original reference and GSAT by Miao et al., 2022 (arxiv 2201.12987, and recently accepted by ICML 2022). Both these two works show that they can fully resolve the distribution shifts discussed in DIR paper. The current results show that DIR fails to achieve effective OOD, which seems to be explained by [6] that DIR failed to establish valid theoretical guarantees for the distribution shifts discussed in DIR paper.
> > > > >
> > > > > Thank you for your suggestions! We will consistently make efforts to make our benchmark easier to use and evaluate new methods. The roadmap of the developments is currently given in [an issue](https://github.com/divelab/GOOD/issues/4) of our GitHub Repo and will be included on the main page as soon as we finish merging the code and dataset modification during the rebuttal period.
> > > > >
> > > > > And we are glad to let you know that the author of [6] from your original reference has contacted us and is engaged in incorporating their method into our benchmark, as https://github.com/divelab/GOOD/issues/4. In terms of the GSAT, we would like to thank you for your reminder. Adding this method is included in our roadmap.
> > > > >
> > > > > Further discussions are welcomed at any time. We would be happy to have insightful discussions with you and make further improvements.
> > > > > GOOD is a growing project. We hope you will keep watching our future updates!
> > > > >
> > > > > Sincerely,
> > > > >
> > > > > Authors

---

### Official Review · Reviewer_R994 · 2022-07-21
**A Well-written Paper but Motivation Can be Improved**

**Rating:** 6
**Confidence:** 3

**Strengths:**

This benchmark, to the best of my knowledge, is the first one to focus on graph out-of-distribution learning tasks. The datasets cover covariate and concept shifts. This could be a new direction in the graph learning domain. The experiments show the effects of out-of-distribution on performance and motivate future works to propose approaches to narrow the performance gap. The datasets are accessible on GitHub and well documented.

**Weaknesses:**

1. One of the major concerns is the motivation for this benchmark. Specifically, there exist several benchmarks for OOD. It is unclear what the difference between the traditional OOD and the OOD on the graph is. This should be explained better because it leads to the motivation of this work. More specifically, what is the role of the graph on the covariate and concept shifts?
2.  It is much better if the authors can compare the difference in data generation for concept shifts and covariate shifts for real-world, semi-artificial, and synthetic datasets.
3. Existing datasets only cover two types of graph learning tasks. I am happy to see more tasks involved such as link prediction.

**Additional Feedback:**

Please reply to the concerns in the above section, and I will raise the score if the authors address my concerns properly.

**Clarity:**

The paper is overall well-written and clearly presented. One quick question is how to choose the training samples and test samples. Since nodes in the graph are not IID. It is unclear whether this may make some difference.

**Correctness:**

The datasets seem not particularly novel. They are derived from existing datasets via different splits of domain selections. But the performance evaluation on the performance gap between ID and OOD is convincing.

**Documentation:**

The authors have provided detailed documents on GitHub, including how to install and use datasets.

**Ethics:**

No ethical issue is found.

**Relation To Prior Work:**

In the related work section, the authors clearly show the differences between their benchmark and existing benchmarks. One quick question is whether they may do the same on other domains rather than graphs. This question is related to the unclear motivation of this work.

**Summary And Contributions:**

This paper proposes a new benchmark GOOD, which focuses on the graph out-of-distribution learning task, where the distribution of the training set is different from that of the test set. Specifically, the benchmark consists of 8 datasets and 14 domain selections on the node classification task and the graph classification task. The authors consider both covariate and concept shifts and obtain 42 different splits. The experiments demonstrate significant performance gaps between in-distribution and out-of-distribution settings.

---

> ### Author Response · Authors · 2022-08-21
> **Major revision based on your suggestions has been made. (Part I)**
>
> Dear reviewer R994,
>
> Thank you for your comprehensive comments and constructive suggestions, which help to improve our work a lot! We have conducted substantial experiments and also revised the paper heavily. We also provide responses for each concern here.
>
>
> > 1. One of the major concerns is the motivation for this benchmark. Specifically, there exist several benchmarks for OOD. It is unclear what the difference between the traditional OOD and the OOD on the graph is. This should be explained better because it leads to the motivation of this work. More specifically, what is the role of the graph on the covariate and concept shifts?
>
> - Thank you for this comment! It is a very useful suggestion to us. We clarify this concern below, which has also been added to the introduction in our revised paper.
>
> - We included the unique challenges and distinctions of OOD generalization on graphs in Section 1 of the revised paper. Traditional OOD methods commonly focus on simple structure equation models [1, 2, 3, 4] or computer vision tasks [4, 5, 6, 7]. In these tasks, the inputs are variables or image features, denoted as $F$. However, graph data possess the complex nature of irregularity and connectivity in topology. A graph is commonly represented as an input pair $(F, A)$, where $F$ are node features, and $A$ is the adjacency matrix. Consequently, **graph OOD problem focuses not only on general feature distribution shifts but also on structure distribution shifts**. Graph neural networks are designed for $F$ and $A$ to pass messages with different strategies, demonstrating that structure and feature comprehensively carry different perspectives of input information of a graph.
> Therefore, in OOD, the particular task field of the graph increase the complexity of covariate and concept shift by including both feature distribution shifts and structure distribution shifts.
> The uniqueness of graph data prompts the development of graph-specific OOD methods [8, 9, 10, 11, 12, 13], and calls for graph OOD benchmarks.
>
>
> > 2. It is much better if the authors can compare the difference in data generation for concept shifts and covariate shifts for real-world, semi-artificial, and synthetic datasets.
>
> - Thank you for your helpful advice! We have added the discussion "Comparison of the covariate(/concept) split design on 3 types of datasets" at the end of Sections 3.1 and 3.2, respectively, where we explain in detail the shift processing on datasets. Please refer to the revised paper for details. We also summarize it below.
>
> - In covariate shift, the difference between synthetic/semi-artificial datasets and real-world datasets is that we can determine the domain of a graph in artificial datasets *arbitrarily* with possible feature modifications. However, the domain of a graph in real-world datasets is selected based on its features attentively.
>
> - The difference of concept splits between (semi-)artificial datasets and real-world datasets locates in whether we can *arbitrarily* determine the concept of a graph. We can define the concept of a graph in (semi-)artificial datasets, but the concept of a graph in real-world datasets is defined with a probability according to the $X_\text{var}$ and target $Y$.
>
> - The difference between the covariate and concept shift in real-world datasets can be directly reflected by the split approaches shown in Sections 3.1 and 3.2. While the covariate shift's environments are only determined by $X$, the concept shift's environments are determined by both $X$ and $Y$.
>
> > 3. Existing datasets only cover two types of graph learning tasks. I am happy to see more tasks involved such as link prediction.
>
> - Many thanks for this constructive suggestion! Adding link prediction datasets is in our project roadmap and will be added as the project develops. To give priority to improving the diversity of real-world datasets, in this update, we added three datasets GOOD-SST2, GOOD-Twitch, and GOOD-WebKB. We schedule link prediction as work after this discussion period. Please keep following the GOOD project to see more updates.
>
> - To explain the reason for our schedule: the datasets of graph/node are expected to be more comprehensive as mentioned by others. We hope to make a high-quality benchmark, thus choosing to improve the real-world diversity of our datasets. It is noteworthy that no existing graph OOD benchmark includes node-level tasks. Therefore, we added one extra graph-level dataset and two node-level datasets to make our benchmark more comprehensive. Furthermore, unlike graph/node prediction, as we know, there are few notable link prediction OOD methods, and the OOD link prediction problem has not been well-defined, currently. We will be making much effort to add the link prediction task and datasets in our next update version of GOOD.
> We hope our justifications and other benchmark improvements are satisfying.

---

> > ### Author Response · Authors · 2022-08-21
> > **Major revision based on your suggestions has been made. (Part II)**
> >
> > > 4. The datasets seem not particularly novel. They are derived from existing datasets via different splits of domain selections. But the performance evaluation on the performance gap between ID and OOD is convincing.
> >
> > - We would like to discuss the novelty of this paper respectfully. In the benchmark track, novelty can be reflected from two perspectives: dataset and benchmark. Though some of our datasets are adapted from prior work, we would like to justify our major novelty from the benchmark perspective.
> > - To the best of our knowledge, **the split process for concept shift** that we proposed in Section 3.2 is first proposed by our paper. Instead of extending the covariate shift to an existing concept shift, we believe that the proposed split method for datasets is innovative.
> > - With the help of the new splitting process, we are the first benchmark that can compare covariate shift and concept shift **on the same domain**, enabling the comparison of different shifts with a proper **variable control**. We also claim that since $P(Y,X)=P(Y|X)P(X)$, the shift consideration of covariate and concept shifts is **complete**, enabling **thorough comparisons of possible shifts on every given domain**. For example, in our benchmark, DIR [9] performs favorably against other algorithms in GOOD-HIV size concept shift while failing in GOOD-HIV size covariate shift. However, no such comparison is possible in other benchmarks to figure out the different behaviors of an algorithm for the same dataset domain.
> > - These contributions are original not only in the field of graph OOD but also in the field of general OOD. These innovations make our GOOD benchmark more comprehensive and systematic than other works.
> > - At last, we believe that **dataset splitting is important** for the OOD benchmark. We are actually trying to simulate the possible real-world OOD situations that may be hard to find now but may be encountered in the future. In the real world, it is extremely difficult to find examples that respectively have the covariate shift and concept shift, given the change of the same variable. Nevertheless, it is possible after the proper splitting process. Since OOD algorithms behave differently for different shifts given the same domain variable, we believe our work is significant in the sense of enabling this comparison.
> >
> > - These contributions are original not only in the field of graph OOD but also in the field of general OOD. With these innovations, our GOOD benchmark is more comprehensive and systematic than other works.
> >
> > > 5. One quick question is how to choose the training samples and test samples. Since nodes in the graph are not IID. It is unclear whether this may make some difference.
> >
> > - In node-level tasks, nodes in the big graph are OOD; therefore, we treat each node as input and assign it to train or test sets by a certain domain, which is its own features or the structure around it. This results in the training set and an OOD test set. In graph-level tasks, nodes within a graph are IID, and nodes in different graphs are OOD. We choose the training or test samples according to the domain/concept the graphs belong.
> >
> > > 6. In the related work section, the authors clearly show the differences between their benchmark and existing benchmarks. One quick question is whether they may do the same on other domains rather than graphs. This question is related to the unclear motivation of this work.
> >
> > - We can apply our split methods to other OOD tasks other than graph tasks. The most significant difference is that the domain selection in graph tasks is more complicated due to the graph topology information.
> >
> > We sincerely thank you for your time! Hope we have addressed your concerns through practical efforts. We look forward to your reply and further discussions, thanks!
> >
> > Sincerely,
> >
> > Authors

---

> > > ### Author Response · Authors · 2022-08-21
> > > **References**
> > >
> > > [1] Arjovsky, Martin, et al. "Invariant risk minimization." arXiv preprint arXiv:1907.02893 (2019).
> > >
> > > [2] Ahuja, Kartik, et al. "Invariance principle meets information bottleneck for out-of-distribution generalization." Advances in Neural Information Processing Systems 34 (2021): 3438-3450.
> > >
> > > [3] Rosenfeld, Elan, Pradeep Ravikumar, and Andrej Risteski. "The risks of invariant risk minimization." arXiv preprint arXiv:2010.05761 (2020).
> > >
> > > [4] Lu, Chaochao, et al. "Invariant Causal Representation Learning for Out-of-Distribution Generalization." International Conference on Learning Representations. 2021.
> > >
> > > [5] Tzeng, Eric, et al. "Adversarial discriminative domain adaptation." Proceedings of the IEEE conference on computer vision and pattern recognition. 2017.
> > >
> > > [6] Ganin, Yaroslav, et al. "Domain-adversarial training of neural networks." The journal of machine learning research 17.1 (2016): 2096-2030.
> > >
> > > [7] Sun, Baochen, and Kate Saenko. "Deep coral: Correlation alignment for deep domain adaptation." European conference on computer vision. Springer, Cham, 2016.
> > >
> > > [8] Wu, Y.-X., Wang, X., Zhang, A., He, X., & Chua, T.-S. (2022). DISCOVERING INVARIANT RATIONALES FOR GRAPH NEURAL NETWORKS. ICLR 2022.
> > >
> > > [9] Wu, Q., Zhang, H., Yan, J., & Wipf, D. (2022). HANDLING DISTRIBUTION SHIFTS ON GRAPHS: AN INVARIANCE PERSPECTIVE. ICLR 2022.
> > >
> > > [10] Zhu, Qi, et al. "Shift-robust gnns: Overcoming the limitations of localized graph training data." Advances in Neural Information Processing Systems 34 (2021): 27965-27977.
> > >
> > > [11] Fan, Shaohua, et al. "Generalizing Graph Neural Networks on Out-Of-Distribution Graphs." arXiv preprint arXiv:2111.10657 (2021).
> > >
> > > [12] Bevilacqua, Beatrice, Yangze Zhou, and Bruno Ribeiro. "Size-invariant graph representations for graph classification extrapolations." International Conference on Machine Learning. PMLR, 2021.
> > >
> > > [13] Li, Haoyang, et al. "Ood-gnn: Out-of-distribution generalized graph neural network." IEEE Transactions on Knowledge and Data Engineering (2022).

---

> > > > ### Comment · Reviewer_R994 · 2022-08-22
> > > > **Thank you for the detail rebuttal**
> > > >
> > > > I am overall satisfied with the rebuttal and raise my score to WEAK ACCEPT.

---

> > > > > ### Author Response · Authors · 2022-08-22
> > > > > **Thank you for your reply and any further discussions are welcomed**
> > > > >
> > > > > Dear reviewer R994,
> > > > >
> > > > > Thank you very much for your reply! We are happy to know that your points are well-addressed in the revision and responses. Your valuable comments and suggestions help improve our paper a lot! We would also like to know if there are any other concerns. Further discussions are welcomed at any time. We would be happy to have insightful discussions with you and make further improvements.
> > > > >
> > > > > Sincerely,
> > > > >
> > > > > Authors

---

### Official Review · Reviewer_mxwq · 2022-07-24
**GOOD: A Graph Out-of-Distribution Benchmark**

**Rating:** 6
**Confidence:** 3
**Correctness:** Yes
**Clarity:** Yes, easy to follow.

**Strengths:**

1. The motivation of systemmatically investigating the impacts of different graph OOD scenarios is clearly described. This paper seems to be the first to propose well crafted split design for different shifts easily applied to general datasets.
2. The benchmark covers a wide range of datasets from various domains. The thorough evaluations also provide insightful suggestions such as shift-sensitive model architecture and task-related data augmentations for future graph OOD research.


**Weaknesses:**

1. The math notations in concept shift derivation are slightly abused. To align with the design in Figure 2, the notation in line 122 should be P(Y = y_{j} | X_{inv}, X_{var} = d_i). Subsequently, it is more rigorous to rewrite equation (2) as P_{c_{k}}(Y | X_{inv}, X_{var} = d_{i}) = \sum_{j=1}^{|\mathcal{Y}|} following the definitions stated in line 132.


**Additional Feedback:**

See above.

**Documentation:**

Yes. The documentations in GitHub Repo are sufficient for quick implementation.


**Ethics:**

No ethical issues.

**Relation To Prior Work:**

Yes, the authors provide thorought discussion with related works in the main body.


**Summary And Contributions:**

This paper developed GOOD, an OOD benchmark focusing on covariate and concept shifts for graph data. The authors provide carefully designed data splits to create different shifts and non-trivial performance gap, making it suitable for node and graph classification. Extensive experiments have been conducted on 8 datasets with up to 42 different splits using mutiple baselines such as VREx, GroupDro and Mixup, which demonstrates significant gaps caused by OOD and further sheds light on the future research trend of graph-based OOD topic.

---

> ### Author Response · Authors · 2022-08-21
> **Discussion and explanations are provided to address your concerns.**
>
> Dear reviewer mxwq,
>
> Thanks a lot for your comments and acknowledgment of our novelty! We have conducted substantial experiments and also revised the paper heavily. We provide responses to your concern here.
>
> > 1. The math notations in concept shift derivation are slightly abused. To align with the design in Figure 2, the notation in line 122 should be $P(Y = y_{j} | X_{inv}, X_{var} = d_i)$. Subsequently, it is more rigorous to rewrite equation (2) as $P_{c_{k}}(Y | X_{inv}, X_{var} = d_{i}) = \sum_{j=1}^{|\mathcal{Y}|} $following the definitions stated in line 132.
>
>  Thank you for your suggestion! We have revised 3.1 and 3.2 of the paper, and we would like to clarify two key points.
> - $d_i$ is a domain symbol instead of a specific value for $X_{var}$.
> - The notation in line 122 shows a spurious correlation probability. This probability is a specific value instead of a distribution $P(Y = y_{j} | X_{inv}, X_{var} = x_i)$. This distribution is a function of $X_{inv}$; therefore, we hope to keep this notation for this purpose.
>
> In addition, we would like to present our updated benchmark improvements, including 3 new datasets, 3 new graph-specific methods, and causal explanations added. Please refer to the [Common Response](https://openreview.net/forum?id=8hHg-zs_p-h&noteId=fszCjK8zqx) for details.
>
> We sincerely thank you for your time! Hope we have addressed your concerns through practical efforts. We look forward to your reply and further discussions, thanks!
>
> Sincerely,
>
> Authors

---

> ### Author Response · Authors · 2022-08-25
> **Author's follow up to reviewer mxwq**
>
> Dear Reviewer mxwq,
>
> Thanks again for your valuable comments and suggestions in your initial review, which helps improve our work a lot. Regarding your main concerns on the notations in concept shift, we have revised the paper accordingly and clarified the description in our response. We have also conducted substantial experiments and also revised the paper heavily to make further improvements. Could you please check at your earliest convenience? Thanks!
>
> We hope that you could reply to our response and revision and consider raising your score if we do have addressed your concerns. Also, please let us know if there are any additional concerns or feedback. Thank you very much!
>
> Sincerely,
>
> Authors

---

> ### Author Response · Authors · 2022-08-28
> **Author's follow up to reviewer mxwq one day before end of discussion**
>
> Dear Reviewer mxwq,
>
> Thanks again for your valuable comments and suggestions in your initial review. Regarding your main concerns on the notations in concept shift, we have revised the paper accordingly and clarified the description in our response. We have also conducted substantial experiments and also revised the paper heavily to make further improvements. Could you please check at your earliest convenience? Thanks!
>
> Since the discussion period is approaching the end, we sincerely hope you can let us know if we have addressed your concern and consider raising your score if we do. Also, we welcome any additional discussions or feedback. Thank you!
>
> Sincerely,
>
> Authors

---

### Official Review · Reviewer_SeXB · 2022-07-26

**Rating:** 6
**Confidence:** 4

**Strengths:**

+ The OOD problem is important to the graph learning community.
+ Both covariate and concept shifts are considered.
+ Comprehensive experiments have been done to demonstrate the ID/OOD performance gap and state-of-the-art OOD models performance.

**Weaknesses:**

- This work extends existing datasets (e.g., OGBG-PCBA in WILDS) by additionally considering the concept shift. While I agree that this work makes a comprehensive benchmark on OOD problems, the real-world datasets are all based on very limited domains, namely molecular graphs and citation networks. I still feel the significance of this contribution is a bit incremental.
- Some domain splitting could be further improved to add practical values to the evaluation.
- It seems that the adoption of existing datasets do not strictly follow license of prior works. See my detailed comments in the following sections.

**Additional Feedback:**

1. It could better to consider evolvement over time as another factor of variance. For example, the Pubmed database provides a unique ID of each molecule that is closely related to the year when it is found. OGB also follows this way of splitting molecular graphs.
2. The authors may further justify the selection of degree as domain factors in the Cora and arXiv datasets. Is this of particular semantic meaning in practice?
3. Also, the authors may elaborate on the practical values of using the CMNIST dataset. It appears that very few graph models will cope with such graph datasets converted from images.
4. I do not see any relationship between the baseline models in Section 5.2 and graph learning models. It feels to me Section 5.2 is general to any existing machine learning models and is deviated from the graph domain.


**Clarity:**

The organization of this paper can be improved. Firstly, the definition of covariate and concept shifts should be put in prior to related work, so that readers could avoid possible confusion in the terminology. Secondly, the idea of environments look a bit vague to me. In Section 3.3, the connection of the OOD problem with causal inference models can be elaborated further.

**Correctness:**

Most datasets are adopted from existing datasets. For semi-synthetic and synthetic datasets, as far as I can see they are constructed in a technically sound way.

**Documentation:**

As most datasets are adopted from previous work, the paper does not explicitly provide dataset documentation nor maintenance schedule. It could be better to mention how they collect the original data and do data cleansing or verification.


**Ethics:**

I would like to request for additional ethical reviews. I have concerns over the license of the provided datasets. In the supplementary material, the paper claims that "we closely follow the license rules, which are specified in the papers". The original OGB datasets are MIT licensed (https://github.com/snap-stanford/ogb/blob/master/LICENSE) which requires retaining the copyright notice for derivatives. However, the GOOD datasets do not have the same license notice string and even claim their datasets are GPL licensed, which is a "viral" license that requires every modified version follows the same license. That being said, the GOOD datasets do not strictly follow the license of the upstream datasets and even override them. I wonder whether the authors contact the original dataset creators regarding the rights on modification/derivation of the datasets.

**Relation To Prior Work:**

Most datasets are adopted from prior work. The novelty of this work feels limited to me.

**Summary And Contributions:**

This work presents a comprehensive out-of-domain generalization benchmark for graphs. The proposed GOOD benchmark contains 8 datasets on node/graph tasks with covariate and concept shifts. Moreover, a series of experiments have been conducted to demonstrate the in-distribution and the out-of-distribution performance gap.

---

> ### Author Response · Authors · 2022-08-03
> **Response to Ethics comments by Reviewer SeXB**
>
> Dear reviewer SeXB,
>
> We would like to thank the reviewer for the constructive comments regarding our license details. We sincerely apologize for the confusion on the license. The GOOD code is released with GPL3.0 license, while the GOOD datasets, following the comprehensive license rules of the original datasets we adapted from, are released with MIT license. We apologize for not making statement for the GOOD dataset license correctly and clearly in the paper. We have clarified the license issue in the revised supplementary material, the GitHub repository and the dataset cloud drive. Please check for any further issues and we will surely response to any ethical concerns in the first place.
> We will response to other comments and technical issues in our next response.
> \
> &nbsp;
>
>
> Sincere thanks,
>
> Authors

---

> > ### Comment · Reviewer_uMWP · 2022-08-16
> > **Ethics Review**
> >
> > The ethics concerns regarding the code and data licenses have been addressed by the authors through clarification and revisions in the supplement.

---

> > > ### Author Response · Authors · 2022-08-24
> > > **Thank you for your comments**
> > >
> > > Dear reviewer uMWP,
> > >
> > > Thank you for the ethics review! We appreciate your useful comments.
> > > Please let us know if there are any other concerns.
> > >
> > > Sincerely,
> > >
> > > Authors

---

> ### Author Response · Authors · 2022-08-21
> **Response to reviewer SeXB: thank you and we have added substantial experiments and heavy revision to address all your concerns (Part I)**
>
> Dear reviewer SeXB,
>
> Thank you for your comprehensive comments and constructive suggestions, which helps to improve our work a lot! We have conducted substantial experiments and also revised the paper heavily. We also provide responses for each concern here.
>
> > 1. This work extends existing datasets (e.g., OGBG-PCBA in WILDS) by additionally considering the concept shift. While I agree that this work makes a comprehensive benchmark on OOD problems, the real-world datasets are all based on very limited domains, namely molecular graphs and citation networks. I still feel the significance of this contribution is a bit incremental.
>
> - To substantially address your concern and also coincide with the updated version of GOOD we've been working on, we strive to add three more datasets with totally different domains. More details are presented in the second improvement of the [Common Response](https://openreview.net/forum?id=8hHg-zs_p-h&noteId=fszCjK8zqx) above.
> We have added three real-world datasets with significantly different domains, **GOOD-SST2, GOOD-Twitch, and GOOD-WebKB**. This significantly expands GOOD's domain diversity coverage, including molecules, grammar trees, citation network, gamer network, and webpage linking network, generalizing to broader application scenarios and tasks.
> 1. GOOD-SST2 (adapted from GraphSST2 [6]) is a **graph-level natural language sentiment analysis** dataset, with splits according to the domain "sentence length," implying that sentence length should not be the key to predict sentimental polarity.
> 2. GOOD-Twitch (adapted from [7]) is a **node-level gamer network** dataset. The nodes represent gamers, and node features are games, and the task is to predict whether a user streams mature content. The domain of GOOD-Twitch splits includes "user language", implying that the prediction target should not be biased by the language a user uses.
> 3. GOOD-WebKB [8] is a node-level dataset of **university webpage networks**, in which a node represents a webpage; node features are extracted from bag-of-word; and edges are hyperlinks between webpages. The task of GOOD-WebKB is to predict the classes of webpages. We split GOOD-WebKB according to the domain "university", suggesting that classified webpages are based on the word contents and link connections instead of the university features.
>
> - In addition, to the best of our knowledge, the split method of concept shift is firstly proposed by our paper. This enables the comparisons of different shifts under the same domain, leading to a comprehensive benchmark. Thus we believe that this benchmarking upgrade can be considered as innovative in a dataset and benchmark track. We have also included the novelty justification in Section 1 of the revised paper.
>
> - We continuously make an effort to add to the diversity in our benchmark datasets. Many reachable real-world datasets are not eligible to be modified as OOD datasets. During our dataset searching stage, we realize that most graph-level datasets with **enough domain information** are molecule-related datasets. We deliberately incorporate molecular datasets with **different targets, domains, shifts, and tasks**, including binary classification, multi-label classification, and regression, so they will behave differently in OOD settings.

---

> > ### Author Response · Authors · 2022-08-21
> > **Response to reviewer SeXB (Part II)**
> >
> > > 2. Some domain splitting could be further improved to add practical values to the evaluation.
> >
> > To address this concern, we specifically justify the practical values of each domain selection, which is also included in Section 4 of the revised paper.
> > - For molecular datasets, we design splits based on two domain selections, namely, scaffold and size.
> > The first one is the Bemis-Murcko scaffold which is the two-dimensional carbon backbone of a molecule. It is a widely-used selection of domain [1,2,3] since it is the structural base of a molecule significantly affecting the appearance but seldomly the functionality. The second one is the number of nodes in a molecular graph. The size is commonly discussed as a domain in OOD [4,5] since it is an inevitable structural feature of a graph and easily learned as a spurious source of correlation. Both features, when not determining the label, can become major sources of distribution shifts.
> >
> > - For GOOD-Cora, we generate splits based on two domain selections, word and degree; the former is the word diversity defined by the selected word count of a publication, and the latter is the degree of a node in the graph. Practically, the word diversity, or writing style, is an easy node feature while purely irrelevant to the class.
> > The degree in citation network means popularity in practice, and popularity should not be the cause of the type of paper. As an inevitable structural feature of nodes, it can become a major source of distribution shifts; therefore, this feature is concluded as a part of $X_{var}$.
> > For GOOD-Arxiv, the value behind the selection of time is that training on earlier publications and predicting the newer ones fits the direct application in real-world scenarios, appearing as a typical distribution shift.
> >
> > - The practical values of GOOD-CMNIST can be considered together with the dataset GOOD-Motif. They build a pair of sanity check datasets for graph-level OOD problems. Specifically, GOOD-Motif is a check for **structure shift** with identical node features, while GOOD-CMNIST is designed for **feature shift** with very similar graph structures. Since a graph can be formalized as $(X, A)$ (where $X$ denotes node features; $A$ denotes adjacency matrix), the shift checks for $X$ and $A$ are both important; therefore, the GOOD-CMNIST and GOOD-Motif are generated for this purpose. Another reason for generating GOOD-CMNIST is that CMNIST has become a popular OOD dataset in computer vision. It is interesting and potentially insightful to compare the different model behaviors and performances between images' CMNIST and graphs' CMNIST.
> >
> > > 3. The organization of this paper can be improved. Firstly, the definition of covariate and concept shifts should be put in prior to related work so that readers could avoid possible confusion in the terminology.
> >
> > - Many thanks for this very useful suggestion. We totally agree and modified the paper accordingly. We now present the definition of covariate and concept shifts in Section 1 of the revised paper prior to related work.
> >
> >
> > > 4. Secondly, the idea of environments look a bit vague to me. In Section 3.3, the connection of the OOD problem with causal inference models can be elaborated further.
> >
> > - In our design, **environment** is the unify of **domains** in covariate shift and **concepts** in concept shift. It should be considered as learning auxiliary information. For example, GroupDRO, VREx, and IRM require group/environment information to balance the model performance among environments. DANN and Coral initially target the domain adaption task, which is a special two-environment case of the OOD problem.
> >
> > - Causal inference stems from the basic causality concept, targeting predicting causal effects, and causal learning aims at incorporating causally related inductive bias into deep learning.
> > We agree it is a good idea to elaborate on the connection between OOD problem and causal(/invariant) learning; therefore, we added the discussion of this connection according to your suggestions and reviewer Ytr5's. A detailed discussion and a causal graph have been included in Section 3 of the revised paper. Please refer to the revised paper.

---

> > > ### Author Response · Authors · 2022-08-21
> > > **Response to reviewer SeXB (Part III)**
> > >
> > > > 5. Most datasets are adopted from prior work. The novelty of this work feels limited to me.
> > >
> > > - We would like to respectfully discuss the novelty of this paper. In the benchmark track, novelty can be reflected from two perspectives: dataset and benchmark. Though some of our datasets are adapted from prior work, we would like to justify our major novelty from the benchmark perspective.
> > > - To the best of our knowledge, **the split process for concept shift** that we proposed in Section 3.2 is first proposed by our paper. Instead of extending the covariate shift to an existing concept shift, we believe that the proposed split method for datasets is innovative.
> > > - With the help of the new splitting process, we are the first benchmark that can compare covariate shift and concept shift **on the same domain**, which means we enable the comparison of different shifts with a proper **variable control**. We also claim that since $P(Y,X)=P(Y|X)P(X)$, the shift consideration of covariate and concept shifts is **complete**, enabling **thorough comparisons of possible shifts on every given domain**. For example, in our benchmark, DIR [9] performs favorably against other algorithms in GOOD-HIV size concept shift while fails in GOOD-HIV size covariate shift. However, no such comparison is possible in other benchmarks to figure out the different behaviors of an algorithm for the same dataset domain.
> > > - These contributions are original not only in the field of graph OOD, but also in the field of general OOD. These innovations make our GOOD benchmark more comprehensive and systematic than other works.
> > > - At last, we believe that **dataset splitting is important** for OOD benchmark. We are actually trying to simulate the possible real-world OOD situations that may be hard to find now but may be encountered in the future. In the real world, it is extremely difficult for us to find examples that respectively have the covariate shift and concept shift given the change of the same variable. But it is possible after the proper splitting process. Since OOD algorithms behave differently for different shifts given the same domain variable, we believe our work is significant in the sense of enabling this comparison.
> > >
> > > > 6. As most datasets are adopted from previous work, the paper does not explicitly provide dataset documentation nor maintenance schedule. It could be better to mention how they collect the original data and do data cleansing or verification.
> > >
> > > - Thank you for your suggestions. We have provided dataset documentation as required, including a dataset sheet and descriptions on the GOOD package documentation page, which explicitly describes in multiple blocks and functions the collection of original data and data processing. Please refer to the Docs directed from our GitHub repo. In addition, we further include more data processing details as suggested in the supplementary revision in Appendix A.
> > >
> > > - The detailed maintenance plan is added to the revision in Appendix F. Our maintenance plan can be trusted for several reasons. First, our code is public with all data processing details. Second, our code is well-documented with detailed tutorials and simple-to-use APIs. Third, proper CI and tests are settled for any change of the code, protecting the reproducibility and usability of the datasets and benchmark. This also directly serves for any external contributions and further developments.
> > >
> > > > 7. It could better to consider evolvement over time as another factor of variance. For example, the Pubmed database provides a unique ID of each molecule that is closely related to the year when it is found.
> > >
> > > - Many thanks for this detailed and constructive suggestion! Currently, we involve **time** as a domain selection for GOOD-Arxiv, and we would certainly like to include more diverse datasets on **time**. To give priority to improving the diversity of real-world datasets, in this update, we added three datasets **GOOD-SST2, GOOD-Twitch, and GOOD-WebKB**. More datasets regarding time will be added as the project develops and, please keep following the GOOD project to see more updates.
> > >
> > > > 8. The authors may further justify the selection of degree as domain factors in the Cora and arXiv datasets. Is this of particular semantic meaning in practice?
> > >
> > > - We select the degree of nodes as the domain for GOOD-Cora and GOOD-Arxiv because the degree in citation network means popularity in practice, and popularity should not be the cause of the type of a paper; therefore, this feature is concluded as a part of $X_{var}$. As an inevitable structural feature of nodes, it can become the major source of distribution shifts.

---

> > > > ### Author Response · Authors · 2022-08-21
> > > > **Response to reviewer SeXB (Part IV)**
> > > >
> > > > > 9. Also, the authors may elaborate on the practical values of using the CMNIST dataset. It appears that very few graph models will cope with such graph datasets converted from images.
> > > >
> > > > - The practical values of GOOD-CMNIST can be considered together with the dataset GOOD-Motif. They build a pair of sanity check datasets for graph-level OOD problems. Specifically, GOOD-Motif is a check for **structure shift** with identical node features, while GOOD-CMNIST is designed for **feature shift** with very similar graph structures. Since a graph can be formalized as $(X, A)$ (where $X$ denotes node features; $A$ denotes adjacency matrix), the shift checks for $X$ and $A$ are both important; therefore, the GOOD-CMNIST and GOOD-Motif are generated for this purpose. Another reason for generating GOOD-CMNIST is that CMNIST has become a popular OOD dataset in computer vision. It is interesting and potentially insightful to compare the different model behaviors and performances between images' CMNIST and graphs' CMNIST.
> > > >
> > > > > 10. I do not see any relationship between the baseline models in Section 5.2 and graph learning models. It feels to me Section 5.2 is general to any existing machine learning models and is deviated from the graph domain.
> > > >
> > > > - We consider this additional feedback as a concern. To address this concern and also coincide with the updated version of GOOD we've been working on, we carefully include three more graph-specific OOD algorithms and provide justifications for our selections. The details are elaborated in the first improvement of the [Common Response](https://openreview.net/forum?id=8hHg-zs_p-h&noteId=fszCjK8zqx) above. Including the graph-specific Mixup, we currently have 4 graph-specific algorithms. The introduction of algorithms, performance comparisons, and algorithm supplements of the algorithms are updated in Section 5.2, and we add more graph-specific discussion.
> > > >
> > > > - We have added three other graph-specific OOD algorithms,
> > > > DIR[9], EERM[10], SR-GNN[11], which give GOOD the sufficiency to reflect the performances and capabilities of current graph-specific OOD methods. DIR is an OOD algorithm for graph-classification tasks, while EERM and SRGNN are designed for node-classification tasks.
> > > > - We provide justifications for our selections. We select these three algorithms for 3 reasons.
> > > > These algorithms have passed peer review and have been accepted by major conferences or journals.
> > > > The code of these algorithms is well-maintained; thus, they are reproducible.
> > > > These algorithms target solving the OOD problem on graphs.
> > > > Among all other graph OOD methods within our knowledge, graph OOD algorithms like [12, 13, 14, 15, 16] haven't been accepted by major conferences or journals yet, while the disentangled GNN methods [17, 18, 19] do not target solving the OOD problem. While [20] is an interested and solid paper we would like to implement, the key code for obtaining induced homomorphism densities is missing. The authors claimed to use R-GPM in their paper, but some C++ compile files are missing in R-GPM's GitHub repository, and the API used in the code mismatches with R-GPM's. The missing and mismatching disabled the reproducibility of this algorithm. In conclusion, we believe our selection of graph-specific algorithms is sufficient and of quality.
> > > >
> > > >
> > > > We sincerely thank you for your time! Hope we have addressed your concerns through practical efforts. We look forward to your reply and further discussions, thanks!
> > > >
> > > > Sincerely,
> > > >
> > > > Authors

---

> > > > > ### Author Response · Authors · 2022-08-21
> > > > > **Response to reviewer SeXB (Part V - Reference)**
> > > > >
> > > > > [1] Weihua Hu, Matthias Fey, Marinka Zitnik, Yuxiao Dong, Hongyu Ren, Bowen Liu, Michele Catasta, and Jure Leskovec. Open graph benchmark: Datasets for machine learning on graphs. Advances in neural information processing systems, 33:22118–22133, 2020.
> > > > >
> > > > > [2] Mucong Ding, Kezhi Kong, Jiuhai Chen, John Kirchenbauer, Micah Goldblum, David Wipf, Furong Huang, and Tom Goldstein. A closer look at distribution shifts and out-of-distribution generalization on graphs. In NeurIPS 2021 Workshop on Distribution Shifts: Connecting Methods and Applications, 2021.
> > > > >
> > > > > [3] Yuanfeng Ji, Lu Zhang, Jiaxiang Wu, Bingzhe Wu, Long-Kai Huang, Tingyang Xu, Yu Rong, Lanqing Li, Jie Ren, Ding Xue, et al. DrugOOD: Out-of-distribution (OOD) dataset curator and benchmark for AI-aided drug discovery–a focus on affinity prediction problems with noise annotations. arXiv 2022.
> > > > >
> > > > > [4] Beatrice Bevilacqua, Yangze Zhou, and Bruno Ribeiro. Size-invariant graph representations for graph classiﬁcation extrapolations. ICML 2021.
> > > > >
> > > > > [5] Yingxin Wu, Xiang Wang, An Zhang, Xiangnan He, Tat-Seng Chua. Discovering Invariant Rationales for Graph Neural Networks. ICLR 2022.
> > > > >
> > > > > [6] Yuan, Hao, et al. "Explainability in graph neural networks: A taxonomic survey." arXiv preprint arXiv:2012.15445 (2020).
> > > > >
> > > > > [7] Rozemberczki, Benedek, Carl Allen, and Rik Sarkar. "Multi-scale attributed node embedding." Journal of Complex Networks 9.2 (2021): cnab014.
> > > > >
> > > > > [8] Pei, Hongbin, et al. "Geom-gcn: Geometric graph convolutional networks." arXiv preprint arXiv:2002.05287 (2020).
> > > > >
> > > > > [9] Wu, Y.-X., Wang, X., Zhang, A., He, X., & Chua, T.-S. (2022). DISCOVERING INVARIANT RATIONALES FOR GRAPH NEURAL NETWORKS. ICLR 2022.
> > > > >
> > > > > [10] Wu, Q., Zhang, H., Yan, J., & Wipf, D. (2022). HANDLING DISTRIBUTION SHIFTS ON GRAPHS: AN INVARIANCE PERSPECTIVE. ICLR 2022.
> > > > >
> > > > > [11] Zhu, Qi, et al. "Shift-robust gnns: Overcoming the limitations of localized graph training data." Advances in Neural Information Processing Systems 34 (2021): 27965-27977.
> > > > >
> > > > > [12] Sui, Y., Wang, X., Wu, J., He, X., & Chua, T.-S. (2021). Deconfounded Training for Graph Neural Networks. Arxiv. http://arxiv.org/abs/2112.15089
> > > > >
> > > > > [13] Wang, Xiang, et al. "Deconfounding to Explanation Evaluation in Graph Neural Networks." arXiv preprint arXiv:2201.08802 (2022).
> > > > >
> > > > > [14] Fan, Shaohua, et al. "Generalizing Graph Neural Networks on Out-Of-Distribution Graphs." arXiv preprint arXiv:2111.10657 (2021).
> > > > >
> > > > > [15] Zhang, Shengyu, et al. "Stable Prediction on Graphs with Agnostic Distribution Shift." arXiv preprint arXiv:2110.03865 (2021).
> > > > >
> > > > > [16] Chen, Yongqiang, et al. "Invariance Principle Meets Out-of-Distribution Generalization on Graphs." arXiv preprint arXiv:2202.05441 (2022).
> > > > >
> > > > > [17] Yang, Yiding, et al. "Factorizable graph convolutional networks." Advances in Neural Information Processing Systems 33 (2020): 20286-20296.
> > > > >
> > > > > [18] Ma, Jianxin, et al. "Disentangled graph convolutional networks." International conference on machine learning. PMLR, 2019.
> > > > >
> > > > > [19] Li, Haoyang, et al. "Disentangled contrastive learning on graphs." Advances in Neural Information Processing Systems 34 (2021): 21872-21884.
> > > > >
> > > > > [20] Bevilacqua, Beatrice, Yangze Zhou, and Bruno Ribeiro. "Size-invariant graph representations for graph classification extrapolations." International Conference on Machine Learning. PMLR, 2021.

---

> ### Author Response · Authors · 2022-08-25
> **Author's follow up to reviewer SeXB**
>
> Dear Reviewer SeXB,
>
> Thanks again for your valuable comments and suggestions in your initial review, which helps improve our work a lot. Regarding your main concerns on the novelty of the benchmark and the practical value of domain selections, which we believe is extremely valuable and critical, we have conducted substantial experiments and also revised the paper heavily. We have also clarified all concerns sentence by sentence in our response. Could you please check at your earliest convenience? Thanks!
>
> We hope that you could reply to our response and revision and consider raising your score if we do have addressed your concerns. Also, please let us know if there are any additional concerns or feedback. Thank you very much!
>
> Sincerely,
>
> Authors

---

### Official Review · Reviewer_iege · 2022-07-26
**Good paper but with some potential major mistakes**

**Rating:** 6
**Confidence:** 3
**Correctness:** 1. When P(Y|X) does not change as in …
**Clarity:** Above average.

**Strengths:**

1. Targets the lack of GOOD benchmark, which is a meaningful direction.
2. The explanation of the problem background is pretty clear and is very friendly for people not familiar with this field.

**Weaknesses:**

The analysis on the concept in Section 3.2 seems problematic, which will be detailed in the following.

**Additional Feedback:**

Minor problems:

In line 44, Domainbed -> DomainBed

**Documentation:**

Well documented on Github

**Ethics:**

No ethics issues spotted

**Relation To Prior Work:**

Relation to previous works is discussed.

**Summary And Contributions:**

This paper targets the lack of GOOD benchmarks and curated a GOOD benchmark from different datasets. The motivation is good and the proposed datasets should contribute to the community.

---

> ### Author Response · Authors · 2022-08-21
> **Response to reviewer iege: thank you for your comments and detailed explanations has been provided (Part I)**
>
> Dear reviewer iege,
>
> Thank you for your valuable comments and suggestions, which helps improve our work a lot! We have revised the paper heavily. We also provide responses for each concern here.
>
> > 1. The analysis on the concept in Section 3.2 seems problematic.
>
> We have spent much time improving the explanations of concept shift, and we believe this "concept" is the most tricky and confusing part of our paper. We strive to provide more transparent explanations as the following, which are also added to Section 3.2 of the revised paper.
>
> - Firstly, The key to addressing the confusion is understanding the concept of causality [2] "Causation is not association." Let us denote "A->B" as "A causes B", "A is B's cause", and "B is A's effect". We generally name "->" as a mechanism [1] like a physical mechanism that keeps independent and unchanged in nature. For example, without any other interferences, "if an object in vacuum receives a force (cause: A), then the object will have an acceleration (effect: B)." Therefore, if we denote receiving a force or not as A=1 or 0, have acceleration or not as B=1 or 0, then **the conditional distribution (mechanism) P(B|A) will be invariant**. Notice that a time flow is hidden between the cause and effect; *i.e.*, we have the cause, **then** the effect will happen. Therefore, when we deliberately **intervene** A, B will change accordingly, but when we deliberately change B, A remains unchanged.
>
> - Secondly, let's consider the data generation process [3]. **Everything has its cause, so there must be some causes of the target $Y$ that we denote as $X_{inv}$ here.** Following the same notation in the paper, there are two general data generation processes: 1) $Y\leftarrow X_{inv}\rightarrow X_{var}$, 2) $X_{inv}\rightarrow Y\rightarrow X_{var}$. Note that $\rightarrow$ means causal mechanism here. We consider the interventions in real-world environments. If $X_{var}$ is intervened (it is forced to be another value), $Y$ won't change. However, if $X_{inv}$ is intervened, **then** $Y$ will be changed according to the mechanism between $X_{inv}$ and $Y$ (so does $X_{var}$ in these cases).
>
> **Next, we will respond to the correctness concerns in your comments one by one.**
>
> > 2. In a concept shift scenario, in which P(Y|X) does change, it is not convincing to still assume the existence of such an X_inv that guarantees P(Y|X_inv) doesn't change.
>
> - The current invariant/causal learning community commonly assumes "everything has its cause" when considering the OOD problem, so there must be some causes of the target $Y$ that we can denote as $X_{inv}$. Therefore, we can assume the existence of such an $X_{inv}$ that guarantees $P(Y|X_{inv})$ doesn't change.
> <!-- This belief can be justified intuitively. -->
>
> - This invariant property keeps everywhere not only in covariate shift but also in concept shift.
>
> > 3. Suppose we do find X_inv in the concept shift scenario so that P(Y|X_inv) doesn't change, then what if we deliberately apply the concept change to P(Y|X_inv)?
>
> - Yes, we can do that. However, after this operation, we would observe that $Y$ and $X_{inv}$ lose their causation in nature; therefore, when we deliberately apply a concept change to $P(Y|X_{inv})$ like changing both their values (because we cannot change it by splitting the datasets according to this question's premise), we break the reason in the data and the data will be meaningless. As a result, this operation should have no practical meaning. If we change the value of $X_{inv}$ in a real-world scenario, then $Y$ will be affected accordingly following the invariant $P(Y|X_{inv})$. If they are changed accidentally, for example, by inaccurate observation, then this phenomenon is generally known as noise or part of the uncertainty.
>
> > 4. why cannot P(Y|X) just change for every possible X?
>
> - Yes, $P(Y|X)$ can change for every possible $X$. Since $P(Y|X_{inv})$ will not change as aforementioned, an immediate inference will be that the change can only happen on $P(Y|X_{var})$.

---

> > ### Author Response · Authors · 2022-08-21
> > **Response to reviewer iege (Part II)**
> >
> > > 5. As mentioned in line 95, P(Y|X)=P(Y|X_inv), which means X_var does not change the conditional probability of Y, then why would the shift of P(Y|X) be caused by P(Y|X_var) as mentioned in line 118?
> >
> > - The situations between line 95 and line 118 are different. In line 95 (covariate shift situation), $X_{var}$ is independent with $Y$, so $P(Y|X_{inv}, X_{var})=P(Y|X_{inv})$. In the line 118 (concept shift situation), $X_{var}$ is correlated with $Y$. A example is the second data generation process as aforementioned $X_{inv}\rightarrow Y\rightarrow X_{var}$ where $P(Y|X_{inv})\ne P(Y|X_{inv}, X_{var})$. Since $P(Y|X_{inv})$ won't change, the change can only happen on $P(Y|X_{var})$.
> >
> > - We notice our confusing description in line 95. A clear explanation can be: Since $X_{var}$ is independent to $Y$, $P(Y|X)=P(Y|X_{inv}, X_{var})=P(Y|X_{inv})$. Because $Y$ is fully determined by $X_{inv}$, $P(Y|X_{inv})$ is invariant across domains, guaranteeing that the change of $X_{var}$ won't affect the $P(Y|X)$. Therefore, splitting datasets according to $X_{var}$ will create a covariate shift.
> >
> >
> > > 6. Is there any mathematical difference between the causal correlation and the spurious correlation in terms of the P(Y|X_inv) and P(Y|X_var)?
> >
> > - Yes, take the first data generation process $Y\leftarrow X_{inv}\rightarrow X_{var}$ as an example. $P(Y|X_{inv})$ is a causal correlation or a mechanism because of the existence of $Y\leftarrow X_{inv}$. While $P(Y|X_{var})$ is spurious correlation (confounding association) because the relation between $Y$ and $X_{var}$ is built based on $X_{inv}$ (confounder). Intuitively, the biggest difference between causal association (causation) and other associations is whether the target $Y$ will change if we deliberately **intervene** the value of the source event. In this case, if we intervene $X_{inv}$, $Y$ will change, but if we intervene $X_{var}$, $Y$ remains the same. This intervention operation is introduced as a "do" operation in do-calculus. There are many accessible materials about causality. You can expect more rigorous theories and explanations about these concepts.
> >
> > > 7. Concept shift seems to be same as the commonly known concept drift (I cannot find concept shift anywhere on the Internet), what is the point to change the name?
> >
> > Response: Concept shift generally can be referred as concept drift [5]. In our work, we follow [4] and believe it is reasonable to unify the names of these dataset **shifts** as **covariate shift ($P(X)$), concept shift ($P(Y|X)$), and prior shift ($P(Y)$)** as [4, 6, 7]. Therefore, we can uniformly refer them as **shifts**.
> >
> > > 8. Minor problems: In line 44, Domainbed -> DomainBed
> >
> > - Many thanks for pointing this out. We have corrected this typo.
> >
> > We sincerely thank you for your time! Hope we have addressed your concerns through practical efforts. We look forward to your reply and further discussions, thanks!
> >
> > Sincere,
> >
> > Authors
> >
> > [1] Peters, Jonas, Dominik Janzing, and Bernhard Schölkopf. Elements of causal inference: foundations and learning algorithms. The MIT Press, 2017.
> >
> > [2] Pearl, Judea. Causality. Cambridge university press, 2009.
> >
> > [3] Ahuja, Kartik, et al. "Invariance principle meets information bottleneck for out-of-distribution generalization." Advances in Neural Information Processing Systems 34 (2021): 3438-3450.
> >
> > [4] Moreno-Torres, Jose G., et al. "A unifying view on dataset shift in classification." Pattern recognition 45.1 (2012): 521-530.
> >
> > [5] Widmer, Gerhard, and Miroslav Kubat. "Learning in the presence of concept drift and hidden contexts." Machine learning 23.1 (1996): 69-101.
> >
> > [6] Tian, Junjiao, et al. "Exploring Covariate and Concept Shift for Out-of-Distribution Detection." NeurIPS 2021 Workshop on Distribution Shifts: Connecting Methods and Applications. 2021.
> >
> > [7] Vorburger, Peter, and Abraham Bernstein. "Entropy-based concept shift detection." Sixth International Conference on Data Mining (ICDM'06). IEEE, 2006.

---

> > > ### Comment · Reviewer_iege · 2022-08-27
> > > **Many thanks to the detailed explanations, but I'm still not convinced of the concept shift part**
> > >
> > > Thanks for the detailed explanations for my concerns. The explanations somewhat addressed my concern, but I still think the concept shift part of the work is problematic, i will explains this in detail. In my original review, my concerns are 'Why must there be an X_inv in the concept shift situation, why cannot P(Y|X) just change for every X' and 'Why can X_var (X_ind in the revised version) cause change in P(Y|X)'. After reading the responses, what has been addressed is that I could agree there exists an X_inv that causes Y, if this is a widely adopted idea in the area, but this also makes me realize that actually we cannot achieve meaningful concept shift (by meaningful, I mean to keep the dataset non-trivial, do not break the causation between the X_inv and Y) without changing P(X). In other words, the concept shift idea described in the paper seems invalid, and the dataset splitting does not maintain an unchanged P(X). Taking the colored mnist dataset in the paper as an example, if digit '1' in training set are mostly blue and in testing set are mostly red, then in the feature space (suppose we flatten the data into a vector and the feature space is the vector space, some dimension correspond to the digit and some correspond to color), P(X) of the training set has higher density at the region corresponding to blue '1', while in the testing set, P(X) has higher density at the region correspinding to red '1' but has lower density at the region corresponding to blue '1' (at least lower than in the training set, or there is no concept change). Then the P(X) does not stay unchanged across training and testing sets. Moreover, P(Y|X) can actually stay unchanged if we keep the distribution of feature dimensions for the digit unchanged across training and testing set, since P(Y|X)=P(Y|X_inv) and X_inv could be the digit features. Therefore, the construction of the concept shift splits is same as the covariate shift splits by modifying the X_ind in the data. Also, the example procedure to construct covariate shift and concept shift mentioned in the paper indeed looks similar (not same), both of which manipulate the color of the data (modifying the X in P(X), resulting in changed P(X)). In line 158, it is stated P_ck(X) = P(X) because the input distribution is fixed, but the splitting actually causes the distribution in each split to be different from the original dataset.
> > >
> > > Overall, I still think that this work has great contribution to the community in terms of the datasets and experiments. However, since the concept shift, which is a major idea in designing the experiments, is problematic, I feel that this may be misleading to readers.

---

> > > > ### Author Response · Authors · 2022-08-28
> > > > **Thank you so much for this insightful comment! We have revised the paper to address your concern.**
> > > >
> > > > Dear Reviewer iege,
> > > >
> > > > Thank you so much for your suggestion! It is absolutely an insightful comment you provide! According to what you mentioned, we realize that the introduction of the concept shift can cause misunderstanding.
> > > >
> > > > We have revised section 3.2 to address your concern by modifying descriptions and equations to eliminate the possible misunderstanding. In brief, as we have mentioned in the paper, concept shift contains an inevitable covariate shift, because with $X_{inv}$ causing $Y$, the association built between $Y$ and $X_\text{ind}$ will lead to the change of $P(X_\text{inv}|X_\text{ind})$, so that $P(X)$ will be changed inevitably.
> > > > according to Figure 2(b). In conclusion, covariate shift can occur alone, but concept shift must occur with covariate shift.
> > > >
> > > > In addition, we would like to justify that this change does not affect our experiments and other claims. Our concept shift dataset split process aims at creating strong associations between $X_{ind}$ and $Y$ while originally keeping the unchanged $P(X_{ind})$ will not break the built associations.
> > > >
> > > > Furthermore, even though the concept shift will contain some covariate shift inevitably. The concept shift splits can still measure the influence of strong spurious correlations, while the covariate shift splits cannot. Note that the most significant difference between covariate shift splits and concept shift splits is that the covariate shift split only requires the information of $X$, while the concept shift split makes use of both $X$ and $Y$. In conclusion, admitting that the concept shift splits contains some covariate shifts does not ruin our purpose (benchmarking the algorithms’ performances for different shift preferences) because covariate shift splits and concept shift splits still maintain their most significant shift characteristics.
> > > >
> > > > Besides, we would like to mention the work V-REx [1]. It discusses both covariate shift and concept shift in its section 2.1, and mentions that many works only consider covariate shift, but concept shift and covariate shift can actually co-occur. We believe it also provides some motivation for us to make our benchmark.
> > > >
> > > > Thank you again for your constructive suggestion! And many thanks for your acknowledgment of our contributions! Since the discussion period is approaching the end, we sincerely hope you can let us know if we have addressed your concern and consider raising your score if the revision clears the misunderstanding.
> > > >
> > > > Sincerely,
> > > >
> > > > Authors
> > > >
> > > > [1] David Krueger, Ethan Caballero, Joern-Henrik Jacobsen, Amy Zhang, Jonathan Binas, Dinghuai Zhang, Remi Le Priol, and Aaron Courville. Out-of-distribution generalization via risk extrapolation (REx). In International Conference on Machine Learning, pages 5815–5826. PMLR, 2021.

---

> ### Author Response · Authors · 2022-08-25
> **Author's follow up to reviewer iege**
>
> Dear Reviewer iege,
>
> Thanks again for your valuable comments and suggestions in your initial review, which helps improve our work a lot. Regarding your main concerns on the analysis on concept shift, which we believe is extremely valuable and critical, we have revised the paper heavily to clarify the notations and descriptions in our response. Could you please check at your earliest convenience? Thanks!
>
> We hope that you could reply to our response and revision and consider raising your score if we do have addressed your concerns. Also, please let us know if there are any additional concerns or feedback. Thank you!
>
> Sincerely,
>
> Authors

---

> ### Author Response · Authors · 2022-08-28
> **Author's follow up to reviewer iege one day before end of discussion**
>
> Dear Reviewer iege,
>
> Thanks again for your valuable comments and suggestions, which helps improve our work a lot. Regarding your main concerns on the description of concept shift, which we believe is extremely helpful and critical, we have revised our paper to address your concern. Could you please check at your earliest convenience? Thanks!
>
> Since the discussion period is approaching the end, we will appreciate it if you could let us know whether we have addressed your concerns and consider updating your evaluations accordingly. Thank you!
>
> Sincerely,
>
> Authors

---

> > ### Comment · Reviewer_iege · 2022-08-29
> > **some concerns are successfully addressed, while still needs more revision**
> >
> > Thanks for the extra explanations. The revision indeed got rid of some misleading parts, but there are still some remaining.
> >
> > 1. In line 39 and 141, it is still stated that P(X) is unchanged in the concept shift situation. This is contradict to the revised statement that concept shift would inevitably include covariate shift.
> >
> > 2. The covariate shift setting should also be carefully designed, or it will also contain some concept shift. If the ratios of different labels in each domain are not the same, there would also be spurious correlation between Y and X_ind, and the covariate shift setting would also contain concept shift. Is this already considered in the dataset splitting? I tried to find the corresponding explanations in both paper and Appendix but failed. This point should have been mentioned in paper.
> >
> > The first point can be easily corrected, while the second is still  a concern. If the experimental setting already considered the second point, then I think my concerns are all resolved

---

> > > ### Author Response · Authors · 2022-08-29
> > > **Thank you for your further explanations. We would like to clarify and address the rest of your concern.**
> > >
> > > Dear Reviewer iege:
> > >
> > > Thank you for your reply! We would like to clarify and address the rest of your concern respectfully.
> > >
> > > > In line 39 and 141, it is still stated that P(X) is unchanged in the concept shift situation. This is contradict to the revised statement that concept shift would inevitably include covariate shift.
> > >
> > > Thank you for pointing out. In line 39, we are describing the original statistical definition of concept shift as in the initial works in history [1,2,3], so we kept the formulation from the literature. We have revised the description to clarify this. For line 141, we have made revisions following your suggestions, removing the statement of "P(X) is unchanged" to eliminate possible misunderstandings, since concept shift would inevitably include covariate shift. We also clarified in the revised paper that practically `we can only build major concept shifts with necessary covariate shifts. We will still call it concept shift for simplicity and distinction`, which should make the description clear. We hope this explanation can be satisfying.
> > >
> > > > The covariate shift setting should also be carefully designed, or it will also contain some concept shift. If the ratios of different labels in each domain are not the same, there would also be spurious correlation between Y and X_ind, and the covariate shift setting would also contain concept shift. Is this already considered in the dataset splitting? I tried to find the corresponding explanations in both paper and Appendix but failed. This point should have been mentioned in paper.
> > >
> > > Thank you for the suggestion. According to our causal graph and the definition of $X_{ind}$, $Y$ and $X_{ind}$ are independent; therefore, this guarantees no spurious correlation exists between $Y$ and $X_{ind}$ theoretically. In dataset splitting, the manual selection of $X_{ind}$ has guaranteed the ratios of different labels in each domain should be the same. **Hence, yes, we have considered it in the dataset splitting.**
> > >
> > > We have added this consideration in Appendix A in the revised paper that `covariate shift design does not contain concept shift`.
> > >
> > > Furthermore, we have also discussed the practical situations with the reviewer Ytr5 at [this discussion](https://openreview.net/forum?id=8hHg-zs_p-h&noteId=jpjnSNIavAG) regarding the selection of $X_{ind}$ in practice. Reviewer Ytr5 agreed that his/her concerns have been addressed through detailed discussions, and we have included some discussions at the end of the Section 3's first paragraph.
> > > Therefore, we believe that we have considered your concern in both the theoretical analysis and experimental settings.
> > >
> > > We really appreciate the helpful discussions! Since the discussion period is fast approaching the end, we sincerely hope you can let us know if we have addressed your concern and consider updating your evaluation if the revision clears the misunderstanding.
> > >
> > > Sincerely,
> > >
> > > Authors
> > >
> > > [1] Joaquin Quiñonero-Candela, Masashi Sugiyama, Anton Schwaighofer, and Neil D Lawrence. Dataset shift in machine learning. Mit Press, 2008.
> > >
> > > [2] Jose G Moreno-Torres, Troy Raeder, Rocío Alaiz-Rodríguez, Nitesh V Chawla, and Francisco Herrera. A unifying view on dataset shift in classification. Pattern Recognition, 45(1):521– 530, 2012. ISSN 0031-3203. doi: https://doi.org/10.1016/j.patcog.2011.06.019. URL https: //www.sciencedirect.com/science/article/pii/S0031320311002901.
> > >
> > > [3] Gerhard Widmer and Miroslav Kubat. Learning in the presence of concept drift and hidden contexts. Machine learning, 23(1):69–101, 1996.

---

### Official Review · Reviewer_Cech · 2022-07-27
**Graph OOD benchmark that requires more graph-specific baselines in evaluation**

**Rating:** 6
**Confidence:** 3
**Clarity:** Overall, the paper is written fairly …

**Strengths:**

GOOD provides a graph OOD benchmark to facilitate further research in robustness for graph neural networks. GOOD contains real-world, semi-artificial, and synthetic datasets. Each dataset offers both covariate and concept shift settings, which the authors state is unique among existing OOD benchmarks.

The authors evaluate several OOD methods (including 1 graph OOD specific method) on all datasets.

The authors provide a script to load GOOD datasets and training and evaluation pipelines.


**Weaknesses:**

Though GOOD is proposed as a graph OOD benchmark, the authors only evaluate one graph-OOD specific algorithm in baseline evaluations, Mixup.

The concept shift splits may contain covariate shift.

The real-world datasets do not come from diverse domains (most are molecular prediction tasks).

**Additional Feedback:**

See above.

**Correctness:**

The evaluation methods and experiment design are designed appropriately and performed correctly.

A minor comment: line 47 should be “WILDS [21] collects real-world data from wild” not “… from wilds”



**Documentation:**

Yes.

**Relation To Prior Work:**

Yes. The authors propose several OOD datasets specific to graph neural nets, and they include a few non-synthetic, real-world datasets in their benchmark. They modify existing graph datasets.

**Summary And Contributions:**

This paper addresses the need for graph out-of-distribution datasets and a systematic graph OOD benchmark. They curate 8 datasets: GOOD-HIV (predicting HIV inhibition from a real-world molecular dataset), GOOD-PCBA, GOOD-ZINC (predicting constrained molecular solubility), GOOD-CMNIST, GOOD-Motif, GOOD-Cora (predicting publication type from a citation network), GOOD-Arxiv (predicting subject area from a citation network of CS arXiv papers), and GOOD-CBAS (classifying node types in motif-like graphs).

They provide a total of 14 different domain selections. Each domain selection can be combined with a covariate shift (which they refer to as “diversity shift”), concept shift (“correlation shift”), and no shifts to obtain 42 different splits. They create concept shift splits. On synthetic datasets, they set a domain-output correlation and generate graphs where a specific output is highly correlated with the domain feature according to this correlation.

They evaluate ERM and 6 other major domain generalization baselines (GroupDRO, VREX DeepCORAL, IRM, Mixup, on these datasets), and show that current OOD algorithms do not significantly improve generalization abilities, emphasizing a need for graph-specific OOD algorithms.

---

> ### Author Response · Authors · 2022-08-21
> **Substantially experimental efforts have been made to address your concerns including 3 new datasets and 3 more graph-specific OOD methods (Part I)**
>
> Dear reviewer Cech,
>
> Thank you for your constructive comments! We have made much effort to thoroughly improve our work accordingly and also provide responses for each concern here.
>
> > 1. Though GOOD is proposed as a graph OOD benchmark, the authors only evaluate one graph-OOD specific algorithm in baseline evaluations, Mixup.
>
> - To address this concern, and also coincided with the updated version of GOOD we've been working on, we have **added three other graph-specific OOD algorithms,
> DIR[1], EERM[2], SR-GNN[3],** which give GOOD the sufficiency to reflect performances and capabilities of current graph-specific OOD methods. DIR is an OOD algorithm for graph-classification tasks, while EERM and SRGNN are designed for node-classification tasks.
> - We provide justifications for our selections. We select these three algorithms for 3 reasons.
> These algorithms have passed peer review and have been accepted by major conferences or journals.
> The code of these algorithms is well-maintained; thus, they are reproducible.
> These algorithms target solving the OOD problem on graphs.
> Among all other graph OOD methods within our knowledge, graph OOD algorithms like [4, 5, 6, 7, 8] haven't been accepted by major conferences or journals yet, while the disentangled GNN methods [9, 10, 11] do not target solving the OOD problem. While [12] is an interested and solid paper we would like to implement, the key code for obtaining induced homomorphism densities is missing. The authors claimed to use R-GPM in their paper, but some C++ compile files are missing in R-GPM's GitHub repository, and the API used in the code mismatches with R-GPM's. The missing and mismatching disabled the reproducibility of this algorithm. In conclusion, we believe our selection of graph-specific algorithms is sufficient and of quality.
> - The introduction of algorithms, performance comparisons, and algorithm supplements of the 10 algorithms (4 are graph-specific) GOOD includes are updated in Section 5.2.1, Table 2, and supplementary material of the revised paper.
>
> The details are also elaborated in the first improvement of the [Common Response](https://openreview.net/forum?id=8hHg-zs_p-h&noteId=fszCjK8zqx) above.
>
> > 2. The concept shift splits may contain covariate shift.
>
> - As we addressed in the paper, concept shift splits maybe contain a slight bias which can be addressed as covariate shift. Firstly, here we explain with an example why it is inevitable due to the discrete nature of graphs. Many scaffolds of molecule datasets contain only **one** graph. Therefore, this graph can only be assigned to **one** concept, leading to an inevitable bias, and we can regard this as a slight covariate shift.
> - Secondly, we are confident to say that our designed concept shift dominates the splits and this slight bias is of little effect. We have experimented with adjusting the degree of concept shift. When the degree of this spurious correlation is adjusted, the slight covariate shift caused by the error does not change, while the performance of the model changes significantly, verifying our claim. These biases are acceptable considering the benchmark since the major performance gaps are caused by major concept shifts.

---

> > ### Author Response · Authors · 2022-08-21
> > **Response to reviewer Cech (Part II)**
> >
> > > 3. The real-world datasets do not come from diverse domains (most are molecular prediction tasks).
> >
> > - We have added three real-world datasets with significantly different domains, **GOOD-SST2, GOOD-Twitch, and GOOD-WebKB**. This significantly expands GOOD's domain diversity coverage, including molecules, grammar trees, citation network, gamer network, and webpage linking network, generalizing to broader application scenarios and tasks.
> > 1. GOOD-SST2 (adapted from GraphSST2 [13]) is a **graph-level natural language sentiment analysis** dataset, with splits according to the domain "sentence length," implying that sentence length should not be the key to predict sentimental polarity.
> > 2. GOOD-Twitch (adapted from [14]) is a **node-level gamer network** dataset. The nodes represent gamers, and node features are games, and the task is to predict whether a user streams mature content. The domain of GOOD-Twitch splits includes "user language", implying that the prediction target should not be biased by the language a user uses.
> > 3. GOOD-WebKB [15] is a node-level dataset of **university webpage networks**, in which a node represents a webpage; node features are extracted from bag-of-word; and edges are hyperlinks between webpages. The task of GOOD-WebKB is to predict the classes of webpages. We split GOOD-WebKB according to the domain "university", suggesting that classified webpages are based on the word contents and link connections instead of the university features.
> >
> > More details are presented in the second improvement of the [Common Response](https://openreview.net/forum?id=8hHg-zs_p-h&noteId=fszCjK8zqx) above.
> >
> > - It is worth mentioning that most reachable real-world datasets are not eligible to be modified as OOD datasets. During our dataset searching stage, we realize that most graph-level datasets with **enough domain information** are molecule-related datasets. With the three molecular datasets, we deliberately incorporate molecular datasets with different targets and tasks to our best, including binary classification, multi-label classification, and regression, so they will behave differently in OOD settings.
> >
> > We sincerely thank you for your time! Hope we have addressed your concerns through practical efforts. We look forward to your reply and further discussions, thanks!
> >
> > Sincerely,
> >
> > Authors
> >
> > [1] Wu, Y.-X., Wang, X., Zhang, A., He, X., & Chua, T.-S. (2022). DISCOVERING INVARIANT RATIONALES FOR GRAPH NEURAL NETWORKS. ICLR 2022.
> >
> > [2] Wu, Q., Zhang, H., Yan, J., & Wipf, D. (2022). HANDLING DISTRIBUTION SHIFTS ON GRAPHS: AN INVARIANCE PERSPECTIVE. ICLR 2022.
> >
> > [3] Zhu, Qi, et al. "Shift-robust gnns: Overcoming the limitations of localized graph training data." Advances in Neural Information Processing Systems 34 (2021): 27965-27977.
> >
> > [4] Sui, Y., Wang, X., Wu, J., He, X., & Chua, T.-S. (2021). Deconfounded Training for Graph Neural Networks. Arxiv. http://arxiv.org/abs/2112.15089
> >
> > [5] Wang, Xiang, et al. "Deconfounding to Explanation Evaluation in Graph Neural Networks." arXiv preprint arXiv:2201.08802 (2022).
> >
> > [6] Fan, Shaohua, et al. "Generalizing Graph Neural Networks on Out-Of-Distribution Graphs." arXiv preprint arXiv:2111.10657 (2021).
> >
> > [7] Zhang, Shengyu, et al. "Stable Prediction on Graphs with Agnostic Distribution Shift." arXiv preprint arXiv:2110.03865 (2021).
> >
> > [8] Chen, Yongqiang, et al. "Invariance Principle Meets Out-of-Distribution Generalization on Graphs." arXiv preprint arXiv:2202.05441 (2022).
> >
> > [9] Yang, Yiding, et al. "Factorizable graph convolutional networks." Advances in Neural Information Processing Systems 33 (2020): 20286-20296.
> >
> > [10] Ma, Jianxin, et al. "Disentangled graph convolutional networks." International conference on machine learning. PMLR, 2019.
> >
> > [11] Li, Haoyang, et al. "Disentangled contrastive learning on graphs." Advances in Neural Information Processing Systems 34 (2021): 21872-21884.
> >
> > [12] Bevilacqua, Beatrice, Yangze Zhou, and Bruno Ribeiro. "Size-invariant graph representations for graph classification extrapolations." International Conference on Machine Learning. PMLR, 2021.
> >
> > [13] Yuan, Hao, et al. "Explainability in graph neural networks: A taxonomic survey." arXiv preprint arXiv:2012.15445 (2020).
> >
> > [14] Rozemberczki, Benedek, Carl Allen, and Rik Sarkar. "Multi-scale attributed node embedding." Journal of Complex Networks 9.2 (2021): cnab014.
> >
> > [15] Pei, Hongbin, et al. "Geom-gcn: Geometric graph convolutional networks." arXiv preprint arXiv:2002.05287 (2020).

---

> ### Author Response · Authors · 2022-08-25
> **Author's follow up to reviewer Cech**
>
> Dear Reviewer Cech,
>
> Thanks again for your valuable comments and suggestions in your initial review, which helps improve our work a lot. Regarding your main concerns on more diverse dataset domain and more graph OOD algorithms, which we believe is extremely valuable and critical, we have conducted substantial experiments and also revised the paper heavily in our response. Could you please check at your earliest convenience? Thanks!
>
> We hope that you could reply to our response and revision and consider raising your score if we do have addressed your concerns. Also, please let us know if there are any additional concerns or feedback. Thank you!
>
> Sincerely,
>
> Authors

---

> ### Author Response · Authors · 2022-08-28
> **Author's follow up to reviewer Cech one day before end of discussion**
>
> Dear Reviewer Cech,
>
> Thanks again for your valuable comments and suggestions in your initial review, which helps improve our work a lot. Regarding your main concerns on more diverse dataset domain and more graph OOD algorithms, which we believe is extremely valuable and critical, we have conducted substantial experiments and also revised the paper heavily in our response. Could you please check at your earliest convenience? Thanks!
>
> Since the discussion period is approaching the end, we sincerely hope you can let us know if we have addressed your concern and consider raising your score if we do. Also, please let us know if there are any additional concerns or feedback. Thank you!
>
> Sincerely,
>
> Authors

---

> > ### Comment · Reviewer_Cech · 2022-08-28
> > **Detailed rebuttal and satisfactory addition of diverse graph datasets and graph-specific OOD baselines**
> >
> > Thank you for your detailed rebuttal and additional experiments.
> >
> > I am satisfied with the authors' additions of 3 graph datasets from diverse/non-molecular domains and 3 graph-specific OOD baselines. I raise my score to 6: weak accept.

---

> > > ### Author Response · Authors · 2022-08-28
> > > **Thank you for your acknowledgment and any further discussions are welcomed**
> > >
> > > Dear reviewer Cech,
> > >
> > > Thank you very much for your reply! We are happy to know that your points are well-addressed in the revision and responses. Your valuable comments and suggestions help improve our paper a lot! Further discussions are welcomed at any time. We would be happy to have insightful discussions with you and make further improvements.
> > >
> > > Sincerely,
> > >
> > > Authors

---

### Review · Ethics_Reviewer_mZRq · 2022-08-21

**Recommendation:** 1

**Ethics Documentation:**

The authors present several real-world, semi-artificial, and synthetic datasets. They describe the process for creating the synthetic and semi-artificial data in the text. For real-world data, the authors use public datasets and have a license statement included. It is not immediately obvious to me where each dataset is coming from -- they cite several papers which I assume created these datasets, but which datasets they are using from each paper is unclear.  The authors have included a statement on maintenance in the supplement and plan to respond to requests, etc. on their GitHub page. The documentation for working with the datasets is clear.

**Ethics Review:**

The ethics concerns regarding the code and data licenses have been addressed by the authors through clarification and revisions in the supplement. Additionally, the they have added several datasets in order to diversify the domains and datasets.

---

> ### Author Response · Authors · 2022-08-24
> **Thank you for the comments**
>
> Dear reviewer mZRq,
>
> Thank you for the ethics review! We appreciate your useful comments.
>
> We have included the introduction and data source of each dataset in the first sentence of each paragraph in Section 4, respectively, along with the corresponding data source citation. Please refer to the revised paper and let us know if there are any other concerns.
>
> Sincerely,
>
> Authors

---

### Author Response · Authors · 2022-08-21
**Common Responses to Questions Raised by Multiple Reviewers (Part I)**

We thank all reviewers for insightful comments and suggestions! Making a comprehensive benchmark for the OOD problem is complicated, and we have been continuously working on the updated version of GOOD after the initial submission. Some of our updates coincide with suggestions from the reviewers. With this progress and adopting all the constructive and solid suggestions, we have made much effort to clear the community's concerns and improve our benchmark thoroughly. Besides providing one-to-one responses, we conclude the significant and commonly-concerned improvements here:

> 1. **Many concerns about `the lack of graph-specific OOD algorithms` (from reviewers Cech, SeXB, and Ytr5).**

We have added three graph-specific algorithms, namely, DIR[1], EERM[2], SR-GNN[3], which give GOOD the sufficiency to reflect the performances and capabilities of current graph-specific OOD methods. **DIR is an OOD algorithm for graph-classification tasks, while EERM and SRGNN are designed for node-classification tasks.**

We select these three algorithms for 3 reasons.
1. These algorithms have passed peer review and have been accepted by major conferences or journals.
2. The code of these algorithms is well-maintained; thus, they are reproducible.
3. These algorithms target solving the OOD problem on graphs.

Among all other graph OOD methods within our knowledge, graph OOD algorithms like [4, 5, 6, 7, 8] haven't been accepted by major conferences or journals yet, while the disentangled GNN methods [9, 10, 11] do not target solving the OOD problem. While [12] is an interested and solid paper we would like to implement, the key code for obtaining induced homomorphism densities is missing. The authors claimed to use R-GPM in their paper, but some C++ compile files are missing in R-GPM's GitHub repository, and the API used in the code mismatches with R-GPM's. The missing and mismatching disabled the reproducibility of this algorithm. **In conclusion, we believe our selection of graph-specific algorithms is sufficient and of quality.**

**In total, GOOD contains 10 algorithms, among which 4 are graph-specific. The introduction of algorithms, performance comparisons, and algorithm supplements are updated in `Section 5.2.1, Table 2, and supplementary material of the revised paper`.**

> 2. **More diverse real-world datasets and domains (suggested by reviewer Cech, SeXB).**

We have added three real-world datasets with different domains, **GOOD-SST2, GOOD-Twitch, and GOOD-WebKB**. This significantly expands GOOD's domain diversity coverage, including molecules, grammar trees, citation network, gamer network, and webpage linking network, generalizing to broader application scenarios and tasks.
- GOOD-SST2 (adapted from GraphSST2 [13]) is a **graph-level natural language sentiment analysis** dataset, with splits according to the domain "sentence length," implying that sentence length should not be the key to predict sentimental polarity.
- GOOD-Twitch (adapted from [14]) is a **node-level gamer network** dataset. The nodes represent gamers, and node features are games, and the task is to predict whether a user streams mature content. The domain of GOOD-Twitch splits includes "user language", implying that the prediction target should not be biased by the language a user uses.
- GOOD-WebKB [15] is a node-level dataset of **university webpage networks**, in which a node represents a webpage. Node features are extracted from bag-of-word, and edges are hyperlinks between webpages. The task of GOOD-WebKB is to predict the classes of webpages. We split GOOD-WebKB according to the domain "university", suggesting that classified webpages are based on the word contents and link connections instead of the university features.

**The introduction, meta info, split details, and supplements of these datasets are added to `Section 4 of the revised paper`.**

---

> ### Author Response · Authors · 2022-08-21
> **Common Responses to Questions Raised by Multiple Reviewers (Part II)**
>
> > 3. **Novelty and contributions of this paper**
>
> - To the best of our knowledge, **the split process for concept shift** that we proposed in Section 3.2 is first proposed by our paper. Instead of extending the covariate shift to an existing concept shift, we believe that the proposed split method for datasets is innovative.
> - With the help of the new splitting process, we are the first benchmark that can compare covariate shift and concept shift **on the same domain**, which means we enable the comparison of different shifts with a proper **variable control**. We also claim that since $P(Y,X)=P(Y|X)P(X)$, the shift consideration of covariate and concept shifts is **complete**, enabling **thorough comparisons of possible shifts on every given domain**. For example, in our benchmark, DIR [9] performs favorably against other algorithms in GOOD-HIV size concept shift while fails in GOOD-HIV size covariate shift. However, no such comparison is possible in other benchmarks to figure out the different behaviors of an algorithm for the same dataset domain.
> - These contributions are original not only in the field of graph OOD, but also in the field of general OOD. These innovations make our GOOD benchmark more comprehensive and systematic than other works.
> - At last, we believe that **dataset splitting is important** for OOD benchmark. We are actually trying to simulate the possible real-world OOD situations that may be hard to find now but may be encountered in the future. In the real world, it is extremely difficult for us to find examples that respectively have the covariate shift and concept shift given the change of the same variable. But it is possible after the proper splitting process. Since OOD algorithms behave differently for different shifts given the same domain variable, we believe our work is significant in the sense of enabling this comparison.
>
> [1] Wu, Y.-X., Wang, X., Zhang, A., He, X., & Chua, T.-S. (2022). DISCOVERING INVARIANT RATIONALES FOR GRAPH NEURAL NETWORKS. ICLR 2022.
>
> [2] Wu, Q., Zhang, H., Yan, J., & Wipf, D. (2022). HANDLING DISTRIBUTION SHIFTS ON GRAPHS: AN INVARIANCE PERSPECTIVE. ICLR 2022.
>
> [3] Zhu, Qi, et al. "Shift-robust gnns: Overcoming the limitations of localized graph training data." Advances in Neural Information Processing Systems 34 (2021): 27965-27977.
>
> [4] Sui, Y., Wang, X., Wu, J., He, X., & Chua, T.-S. (2021). Deconfounded Training for Graph Neural Networks. Arxiv. http://arxiv.org/abs/2112.15089
>
> [5] Wang, Xiang, et al. "Deconfounding to Explanation Evaluation in Graph Neural Networks." arXiv preprint arXiv:2201.08802 (2022).
>
> [6] Fan, Shaohua, et al. "Generalizing Graph Neural Networks on Out-Of-Distribution Graphs." arXiv preprint arXiv:2111.10657 (2021).
>
> [7] Zhang, Shengyu, et al. "Stable Prediction on Graphs with Agnostic Distribution Shift." arXiv preprint arXiv:2110.03865 (2021).
>
> [8] Chen, Yongqiang, et al. "Invariance Principle Meets Out-of-Distribution Generalization on Graphs." arXiv preprint arXiv:2202.05441 (2022).
>
> [9] Yang, Yiding, et al. "Factorizable graph convolutional networks." Advances in Neural Information Processing Systems 33 (2020): 20286-20296.
>
> [10] Ma, Jianxin, et al. "Disentangled graph convolutional networks." International conference on machine learning. PMLR, 2019.
>
> [11] Li, Haoyang, et al. "Disentangled contrastive learning on graphs." Advances in Neural Information Processing Systems 34 (2021): 21872-21884.
>
> [12] Bevilacqua, Beatrice, Yangze Zhou, and Bruno Ribeiro. "Size-invariant graph representations for graph classification extrapolations." International Conference on Machine Learning. PMLR, 2021.
>
> [13] Yuan, Hao, et al. "Explainability in graph neural networks: A taxonomic survey." arXiv preprint arXiv:2012.15445 (2020).
>
> [14] Rozemberczki, Benedek, Carl Allen, and Rik Sarkar. "Multi-scale attributed node embedding." Journal of Complex Networks 9.2 (2021): cnab014.
>
> [15] Pei, Hongbin, et al. "Geom-gcn: Geometric graph convolutional networks." arXiv preprint arXiv:2002.05287 (2020).

---

### Meta-Review · Area_Chair_PEqe · 2022-09-08

**Recommendation:** Accept
**Confidence:** 4

**Metareview:**

This paper proposed a benchmark for out-of-distribution generalization in graph data. The problem is interesting and the authors made a lot of efforts in addressing the reviewers' comments. I do think the authors addressed most of the reviewers' concerns. In addition, graph OOD benchmark can indeed benefit the community, which is still underexplored in existing benchmarks, such as WILDS, MetaShifts, NICO, and Shifts.

---

### Decision · Program_Chairs · 2022-09-16

Accept